# Genome-wide association studies in a large Korean cohort identify quantitative trait loci for 36 traits and illuminate their genetic architectures

Yon Ho Jee [1], Ying Wang [2,3], Keum Ji Jung [4] ✉, Ji-Young Lee[4], Heejin Kimm[4], Rui Duan [5], Alkes L. Price [1,5,6], Alicia R. Martin [2,3,7] & Peter Kraft [1,8] ✉

Genome-wide association studies (GWAS) have predominantly focused on European ancestry populations, limiting biological discoveries across diverse populations. Here we report GWAS findings from 153,950 individuals across 36 quantitative traits in the Korean Cancer Prevention Study-II (KCPS2) Biobank. We discovered 301 previously unreported genetic loci in KCPS2, including an association between thyroid-stimulating hormone and *CD36*. Meta-analysis with the Korean Genome and Epidemiology Study, Biobank Japan, Taiwan Biobank, and UK Biobank identified 4588 loci that were not significant in any contributing GWAS. We describe differences in genetic architectures across these East Asian and European samples. We also highlight East Asian specific associations, including a known pleiotropic missense variant in *ALDH2*, which fine-mapping identified as a likely causal variant for multiple traits. Our findings provide insights into the genetic architecture of complex traits in East Asian populations and highlight how broadening the population diversity of GWAS samples can aid discovery.

Large-scale biobanks integrating genomic and electronic health record data enable genome-wide association studies (GWAS) to identify numerous genetic associations and provide insights into the biological mechanisms of human complex traits and diseases[1,2]. In turn, the combined effects of these genetic markers can be summarized as polygenic risk score to estimate individuals' genetic predispositions for complex diseases, which have successfully identified individuals with a high risk of disease[3,4]. However, current genetic discovery efforts heavily underrepresent non-European populations globally and thus limit further discoveries of variants that are rare or absent in European (EUR) populations but common in other ancestry groups[5]. Furthermore, this genomic research imbalance could lead to health disparities if the genomic discoveries benefit only European ancestry individuals in clinical practice[6].

Early efforts toward diversifying GWAS in East Asian (EAS) populations, including Biobank Japan (BBJ)[7–9], Korean Genome and Epidemiology Study (KoGES)[10–13], and China Kadoorie Biobank[14], Taiwan Biobank (TWB)[15] have made significant

[1]Department of Epidemiology, Harvard T.H. Chan School of Public Health, Boston, MA, USA. [2]Analytic and Translational Genetics Unit, Massachusetts General Hospital, Boston, MA, USA. [3]Stanley Center for Psychiatric Research and Program in Medical and Population Genetics, Broad Institute of MIT and Harvard, Cambridge, MA, USA. [4]Institute for Health Promotion, Department of Epidemiology and Health Promotion, Graduate School of Public Health, Yonsei University, Seoul, Korea. [5]Department of Biostatistics, Harvard T.H. Chan School of Public Health, Boston, MA, USA. [6]Program in Medical and Population Genetics, Broad Institute of MIT and Harvard, Cambridge, MA, USA. [7]Department of Medicine, Harvard Medical School, Boston, MA, USA. [8]Trans-Divisional Research Program, Division of Cancer Epidemiology and Genetics, National Cancer Institute, National Institutes of Health, Boston, MD, USA. ✉e-mail: KJJUNG@yuhs.ac; phillip.kraft@nih.gov

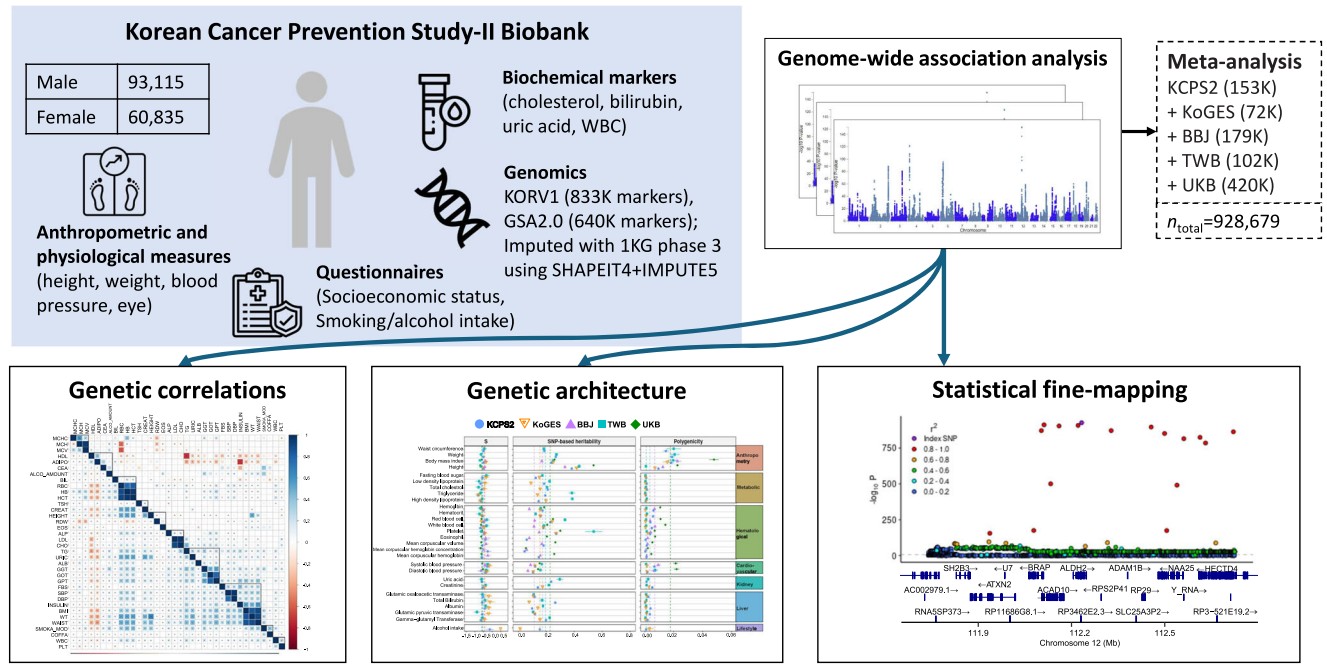

**Fig. 1 | Overview of the Korean Cancer Prevention Study-II Biobank and analysis.** Detailed descriptions of the 36 quantitative traits examined in this study are shown in Supplementary Data 1. After QC, the data were phased using SHAPEIT4[71] and imputed using IMPUTE5[72] with 1000 Genomes Project Phase 3 data.

contributions and facilitated various genetic studies in these populations. Despite these efforts, the representation of EAS groups in genetic research remains low, compared to European groups (e.g., UK Biobank [UKB][16], FinnGen[17], HUNT[18], and deCODE[19]). For example, one of the most extensively used resources is UKB, which includes approximately 500,000 British individuals with deep phenotyping and genomic data[20]. According to the GWAS Diversity Monitor[21], over 90% of total GWAS participants are from European-ancestry samples, while only 4% of participants are of Asian origin despite making up 59% of the global population. The inclusion of additional EAS biobanks is warranted to empower genetic discovery and elucidate the genetic architecture of complex traits and diseases within East Asia.

Here we conducted GWAS for 36 quantitative traits from 153,950 individuals in the Korean Cancer Prevention Study-II (KCPS-II) Biobank[22], a prospective cohort study of the Korean population with genomic data and a wide range of measured phenotypes. Following the GWAS in KCPS2, we meta-analyzed 21 traits across KCPS2, KoGES, BBJ, TWB, and UKB to identify significant loci across East Asian and European ancestry populations. We compared the genetic architectures of these traits across populations leveraging GWAS summary statistics from KCPS2, KoGES, BBJ, TWB, and UKB. Lastly, we pinpointed putatively causal variants through fine-mapping and conducted colocalization to understand the biological mechanisms underlying these traits.

## Results

A total of 153,950 participants were genotyped, including 64,812 participants on the GSA-chip array and 89,138 participants on the Korean-chip array in this study. We subsequently conducted genotype quality control (QC) and imputation. Figure 1 provides an overview of the KCPS2 samples, the traits examined, their abbreviations, and the analyses conducted in this study (Supplementary Data 1). We analyzed 36 quantitative traits including 4 anthropometric traits, 7 metabolic biomarkers, 5 liver function enzymes, 1 thyroid hormone, 1 tumor marker, 3 kidney function traits, 10 hematological traits, 2 cardiovascular traits, and 3 lifestyle factors.

### GWAS of KCPS2 and pleiotropy analysis
We conducted GWAS of 36 human quantitative traits in the KCPS2 Biobank ($n = 153,950$). We used a linear mixed model implemented in SAIGE[23] for association testing to maximize statistical power and included age, sex, 10 principal components (PCs), and SNP array as covariates. None of the GWAS exhibited striking systematic inflation in test statistics indicative of population stratification or other artifact (median $\lambda_{GC}$ 1.23, median S-LDSC intercept 1.04) (Supplementary Data 1). Using S-LDSC with the baseline-LD model, we estimated the SNP-based heritability for each trait (Supplementary Data 1), which ranged from 0.034 (alcohol intake) to 0.347 (height).

Our analysis discovered 2962 independent genome-wide significant loci (median 68, range 1–428 loci; 2631 unique loci) across 36 traits using the 1000 Genome phase 3 EAS samples as the LD reference (Supplementary Data 2). Among these, 301 loci (median 6, range 0–32 loci) were not reported in previous GWAS[24] related to the corresponding trait using Experimental Factor Ontology (EFO) term (Fig. 2a, Supplementary Data 3), with the greatest fraction of novel loci (novelty rate) for carcinoembryonic antigen [CEA, 10/16 (63%) novel loci], followed by thyroid-stimulating hormone [TSH, 32/100 (32%) novel loci]. (The operational definition of "novel association" is detailed in the Methods.) Across 21 traits available in all five biobanks, 36% of the novel loci replicated in a meta-analysis of KoGES, BBJ and TWB (Supplementary Data 12). The novel loci tend to be more common in KCPS2 than in 1000 Genome phase 3 EUR samples (median KCPS2 minor allele frequencies [MAF]: 0.207 vs. median EUR MAF: 0.118; paired $t$ test $P = 2.2e-16$ vs. median EAS MAF: 0.202; paired $t$-test $P = 3.8e-4$). (Supplementary Fig. 1). We also identified widespread pleiotropy: 4960 gene regions contained variants associated with one or more traits (mean 2.3 traits, range 1–27). For example, out of 36 traits, variants near ALDH2 were associated with 26 traits, including blood pressure and liver enzyme values (Fig. 2b, Supplementary Data 4).

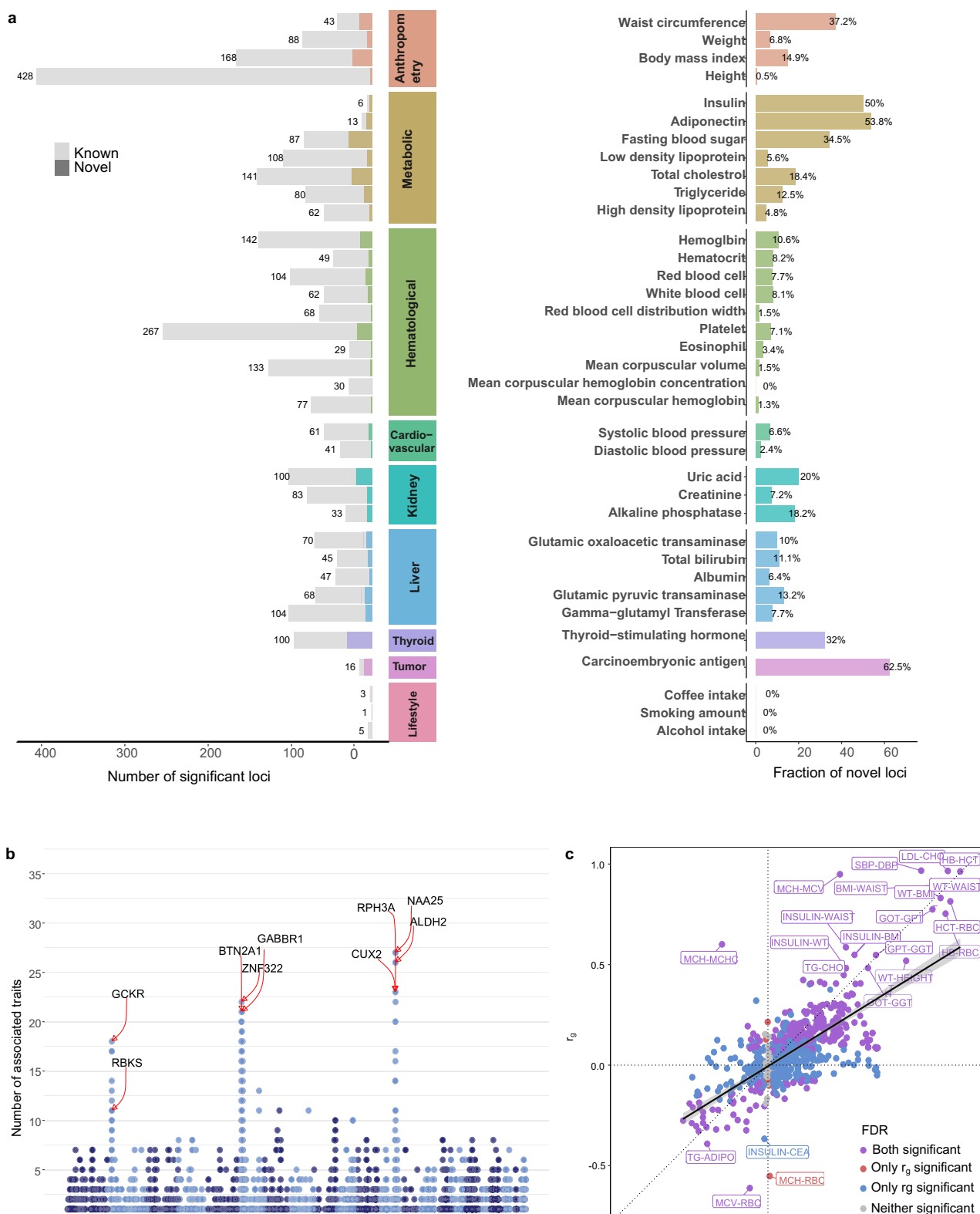

## Genetic and phenotypic correlations between the 36 traits in KCPS2

By estimating pairwise genetic correlations ($r_g$) between traits, we identified clusters of highly genetically correlated traits, including cardiometabolic risk factors (e.g., fasting blood sugar [FBS], systolic blood pressure [SBP], diastolic blood pressure [DBP], insulin, body mass index [BMI], weight, and waist circumference) and liver enzyme traits (e.g., albumin, glutamic oxaloacetic transaminase [GOT], glutamic pyruvic transaminase [GPT], and gamma-glutamyl transferase [GGT]) (Supplementary Data 5, Fig. 2c, Supplementary Fig. 2). The slope of the relationship between pairwise genetic correlations and phenotypic correlations ($r_p$) for 36 traits was 0.634. We identified

**Fig. 2 | GWAS results for 36 quantitative traits in the Korean Cancer Prevention Biobank-II (KCPS2). a** Number of known and novel ("previously unreported") variants identified in KCPS2 compared to the Open Target Genetics[83] using EFO terms (Supplementary Data 2-S3). The definition of 'novel association' is detailed in the Methods. **b** A summary of genome-wide significant loci associated with the 36 traits in KCPS2. Each locus was mapped to a gene using FUMA[77] with a 1000 Genome Phase 3 East Asian reference panel. We then counted the number of associated traits (out of 36 traits) per gene (Supplementary Data 4). (**c**) Comparisons of

pairwise genetic correlations ($r_g$) between phenotypic correlations ($r_p$) for the 36 traits in KCPS2. $r_g$ was estimated using bivariate LDSC based on association test statistics from linear regression. Significant rg and $r_p$ after false discovery rate (FDR < 0.05) correction is indicated by purple if both $r_g$ and $r_p$ were significant, red if only $r_g$ was significant, blue if only $r_p$ was significant, and gray if neither was significant. The black solid line was estimated by spline smoothing from a linear regression model. The complete set of $r_g$ and $r_p$ is available in Supplementary Data 5.

significantly negative genetic and phenotypic correlations of high-density lipoprotein [HDL] cholesterol and adiponectin with the majority of cardiometabolic risk factors including FBS, insulin, and BMI (mean cardiometabolic traits $r_g = -0.24$, $r_p = -0.25$ for HDL; $r_g = -0.27$, $r_p = -0.27$ for adiponectin). These findings are consistent with a known cardioprotective role of HDL[25] and beneficial effects of adiponectin on obesity-associated metabolic and vascular disorders[26,27]. In contrast, the genetic correlations between bilirubin and a number of cardiometabolic risk factors (mean cardiometabolic traits [FBS, insulin, BMI, waist circumference] $r_g = -0.09$), low-density lipoprotein cholesterol ($r_g = -0.13$ [FDR = 0.012]), and WBC ($r_g = -0.11$ [FDR = 0.001]) were significantly negative, although the phenotypic correlations were significantly positive (e.g., mean correlation between bilirubin and cardiometabolic traits $r_p = 0.04$). Bilirubin levels have been shown to be inversely correlated with cardiovascular disease risk by inhibiting cholesterol synthesis and modulating the immune system[28,29], which is supported by our genetic correlations results. Liver enzyme values such as GGT were positively associated with alcohol consumption both genetically and phenotypically ($r_g = 0.31$ [FDR = 0.0001], $r_p = 0.33$ [FDR < 0.0001]), consistent with a previous Mendelian randomization (MR) study[30]. Similarly, smoking, alcohol consumption, and hemoglobin showed significantly positive genetic and phenotypic correlations with a number of cardiometabolic risk factors (mean cardiometabolic traits $r_g = 0.18$, $r_p = 0.22$ for smoking; $r_g = 0.14$, $r_p = 0.16$ for alcohol consumption; $r_g = 0.14$, $r_p = 0.28$ for hemoglobin). For hemoglobin, previous MR showed evidence for lower hemoglobin levels being associated with lower BMI, better glucose tolerance and other metabolic profiles, lower inflammatory load, and blood pressure[31].

## Meta-analysis of 21 traits across KCPS2, KoGES, BBJ, TWB, and UKB

We meta-analyzed 21 traits across KCPS2 (153 K), KoGES (72 K), BBJ (179 K), TWB (102 K), and UKB (420 K) and discovered a total of 12,224 loci associated with the 21 traits, among which 4588 were not genome-wide significant in any of the other four contributing GWAS (Fig. 3a, Supplementary Fig. 3, Supplementary Data 6-S7). The median MAF in KCPS2 for the lead variants at the loci which were only significant in the meta-analysis but not significant in the other individual GWAS, was lower than the MAF in KCPS2 for the lead variants at the loci that were only significant in the KCPS (median MAF 0.29 versus 0.31, respectively) (Fig. 3b).

We compared effect sizes from KCPS2 to effect sizes by study for the lead variants at the 12,224 genome-wide significant loci from meta-analysis (Fig. 3c). The correlations between KCPS2 effect sizes and those in other East Asian biobanks are similar (KCPS2-KoGES: 0.890, KCPS2-BBJ: 0.887, KCPS2-TWB: 0.893), with smaller correlation with UKB (KCPS2-UKB: 0.765). To further investigate associations not previously reaching genome-wide significance ($P < 5 \times 10^{-8}$) in KoGES, BBJ, TWB, or UKB GWAS, we compared effect sizes from the meta-analysis to MAF by study (Supplementary Fig. 4). There was an inverse relationship between MAF and effect size, due in part to the restriction to genome-wide significant variants. The lead variants from genome-wide significant loci identified in the meta-analysis had in general similar study-specific MAF in East Asian populations (KCPS2 [median MAF = 0.27], KoGES [median MAF = 0.27], BBJ [median MAF = 0.27], and TWB

[median MAF = 0.28]) compared to European ancestry populations in UKB (median MAF = 0.26).

## Genetic architecture compared between KCPS2, KoGES, BBJ, TWB, and UKB

We investigated the genetic architecture in KCPS2 (Supplementary Fig. 5) and compared it with KoGES, BBJ, TWB, and UKB across seven trait categories (Fig. 4a, Supplementary Data 8). The S parameters linking MAF and effect sizes were similar across the biobanks (median S = −0.51, range -0.80, -0.03), suggesting a pervasive action of negative selection on the trait-associated variants[32]. The SNP-heritability estimates ($h^2_g$) varied widely across different biobanks and categories. For example, compared to BBJ, KCPS2 has higher heritability estimates for anthropometry (median $h^2_g = 0.31$ vs. 0.25), cardiovascular (median $h^2_g = 0.11$ vs. 0.08), and hematological traits (median $h^2_g = 0.18$ vs. 0.11). For hematological traits, UKB has the largest heritability estimates (median $h^2_g = 0.23$), with the exception of platelet ($h^2_g = 0.54$) and RBC ($h^2_g = 0.33$) being the largest in TWB. Compared to TWB, KCPS2 has lower heritability estimates for metabolic (median $h^2_g = 0.18$ vs. 0.22), liver (median $h^2_g = 0.12$ vs. 0.17), and kidney traits (median $h^2_g = 0.19$ vs. 0.26). However, these differences were not statistically significant (Wilcoxon signed-rank test $p > 0.05$), likely due to the limited number of traits being compared (number of paired comparisons range from 1 to 8) (Supplementary Data 14). Overall, we observed high correlations of heritability in KCPS2 with the other biobank (KCPS2-KoGES: Pearson correlation $r = 0.70$, $P = 4e-04$; KCPS2-BBJ: $r = 0.86$, $P = 0.00033$; KCPS2-TWB: $r = 0.80$, $P = 5.2e-06$; and KCPS2-UKB: $r = 0.90$, $P = 7.7e-05$) (Fig. 4b-e). We note that our TWB heritability estimates using SbayesS and unmatched LD were higher than those reported by Chen et al.[15] using LDSC and in-sample LD, especially for metabolic and hematological traits (Supplementary Fig. 6a), which led to a low correlation between the heritability estimates of KCPS2 and TWB ($r = 0.64$, 95% CI: 0.31−0.83, $P = 9.62e-04$) (Supplementary Fig. 6b). When we used the heritability estimates reported by Chen and colleagues, the correlation in heritability between KCPS2 and TWB improved ($r = 0.80$, 95% CI: 0.58−0.91) (Fig. 4d). The most polygenic traits (weight and BMI) had about 2% SNPs with nonzero effects, whereas the least polygenic traits (coffee intake, bilirubin, and MCHC) were affected by about 0.02−0.03% common SNPs in KCPS2. The median polygenicity estimates for the eight traits available in all five studies were largest in UKB (median $\pi = 0.02$), followed by BBJ (median $\pi = 0.007$), KCPS2 (median $\pi = 0.005$), KoGES (median $\pi = 0.002$), and TWB (median $\pi = 0.001$), which in general follows the same order as the sample sizes of the biobanks. Nevertheless, the genetic correlation estimates within EAS were close to 1 (KCPS2-KoGES median $r_g = 0.997$, KCPS2-BBJ median $r_g = 0.885$, KCPS2-TWB median $r_g = 0.926$) and were in general higher than the $r_g$ between EAS and EUR (KCPS2-UKB median $r_g = 0.815$) for these traits (Supplementary Fig. 7, Supplementary Data 9).

## Fine-mapping and colocalization analysis

To identify potential causal variants, we performed a single-population fine-mapping using SuSiE[33] in KCPS2. Specifically, we fine-mapped 26 traits associated with the region spanning *ALDH2* on chromosome 12 (± 500 kb from rs671, which is known to be functionally related to

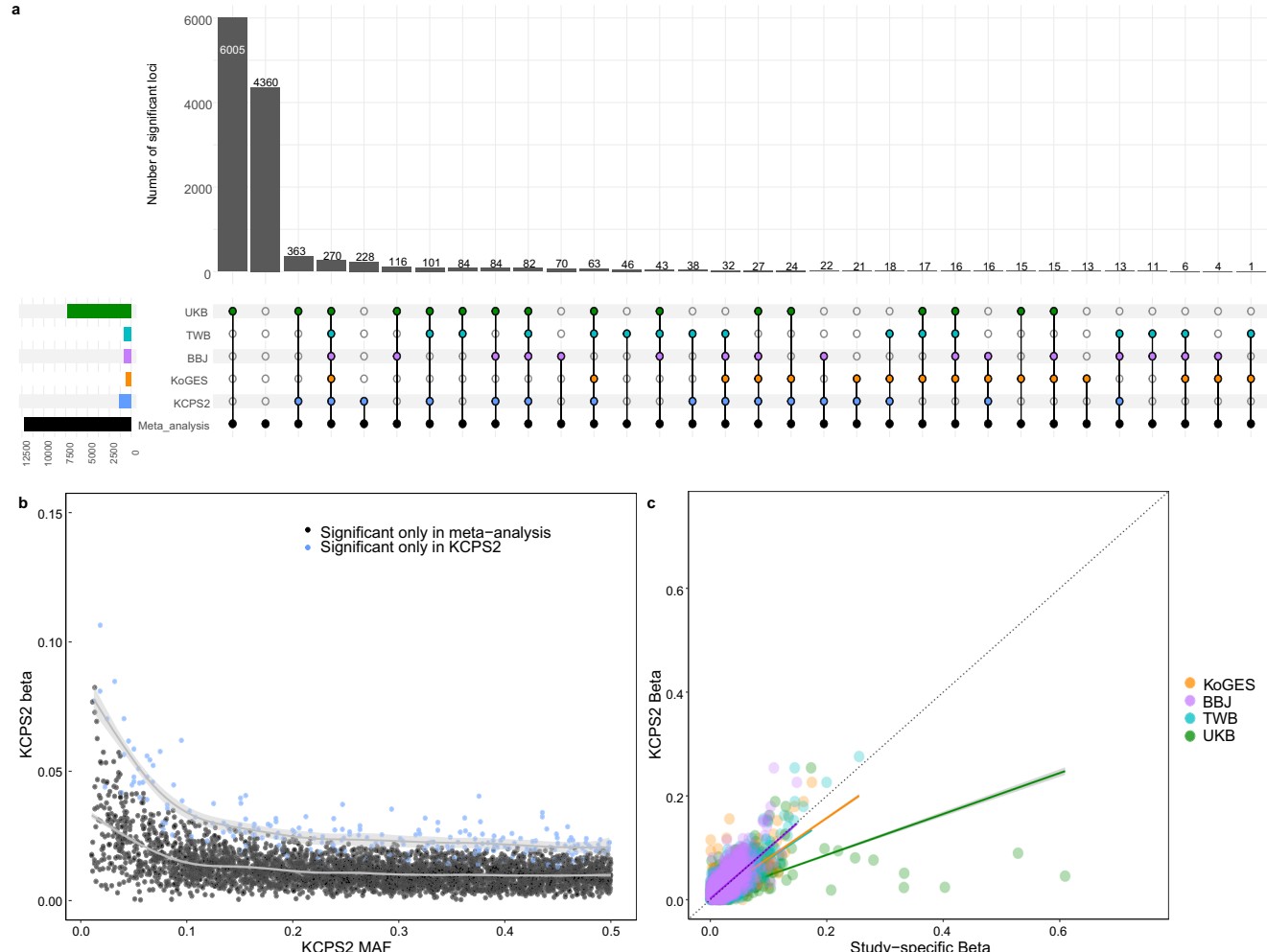

**Fig. 3 | Meta-analysis of 21 traits across KCPS2, KoGES, BBJ, TWB, and UKB.**
**a** Genome-wide significant loci identified in the meta-analysis, Color of dots indicate significance in meta-analysis (black), KCPS2 (blue), KoGES (orange), BBJ (purple), TWB (light blue), and UKB (green). Multiple dots in a bar represent simultaneous significance in multiple cohorts. **b** Comparisons of allele frequency and effect sizes in KCPS2 for the genome-wide significant variants discovered only in KCPS2 (blue) versus those identified only in the meta-analysis (black).

**c** Comparisons of effect sizes in KCPS2 and study-specific effect sizes for the lead variants at the 12,224 meta-analysis genome-wide significant loci. The solid lines were estimated by spline smoothing from generalized additive model (**b**) or linear regression model (**c**). The error bars (**b** and **c**) indicate 95% confidence intervals estimated as ±1.96 × standard error. Full meta-analysis results are shown in Supplementary Data 6, 7.

alcohol metabolism). 1,476 variants in this region were fine-mapped to a total of 56 credible sets, among which 17 contain exactly one variant (median 17.5, range 1–470 variants) (Supplementary Data 10). rs671, a non-synonymous SNP associated with alcohol metabolism and alcohol intake (rs671-A, Beta = -0.59, $P = 1.9 \times 10^{-2658}$), had a posterior inclusion probability (PIP) of greater than 90% for 8 traits including alcohol intake, GOT, GPT, GGT, SBP, DBP, coffee intake, and triglyceride (Fig. 5a, b). For alcohol intake, we found seven credible sets with exactly one variant, all with PIP = 1, including rs671, rs555501971, rs141043717, rs61055528, rs149178839, rs11066008, and rs550463060; in a sensitivity analysis setting the maximum number of credible sets to one (L = 1), only rs671 remained in the credible set (Supplementary Data 11). To assess the sensitivity of these results to analytic approach, we used conditional and joint analysis[34] to conduct association analyses in the *ALDH2* region conditional on rs671 and perform a stepwise selection analysis. These approaches identified independent signals consistent with each other and the original SuSiE analysis (Supplementary Data 10).

To examine the mechanisms underlying these pleiotropic associations, we performed colocalization by pairing GWAS for alcohol intake with GWAS for each of traits where the PIP of rs671 was greater than 90%. These traits included liver enzymes (GOT, GPT, GGT), blood pressure (SBP, DBP), coffee intake, and triglyceride. rs671 was colocalized between alcohol intake and all of these traits with PP4 = 100%, supporting a shared causal variant for these two traits at the locus (Fig. 5b).

## Discussion

In this study, we identified quantitative trait loci for 36 complex traits and investigated the genetic architecture of complex traits in 153,950 Korean individuals. Our analysis discovered 301 previously unreported genetic loci that were not reported in previous GWAS related to the corresponding trait. We also demonstrated widespread pleiotropy and variants near *ALDH2* were associated with 26 traits. Meta-analysis of 21 traits across KCPS2, KoGES, BBJ, TWB, and UKB identified 4588 loci that were not significant in any of the other four contributing GWAS. We compared the genetic architecture of these traits in KCPS2, KoGES, BBJ, TWB, and UKB, and pinpointed one of the most pleiotropic regions (*ALDH2*) through fine-mapping, which colocalized with high probability with a diverse set of traits such as liver enzyme values.

Our study underscores the importance of enhancing the ancestral diversity and sample size of GWAS samples to facilitate genetic

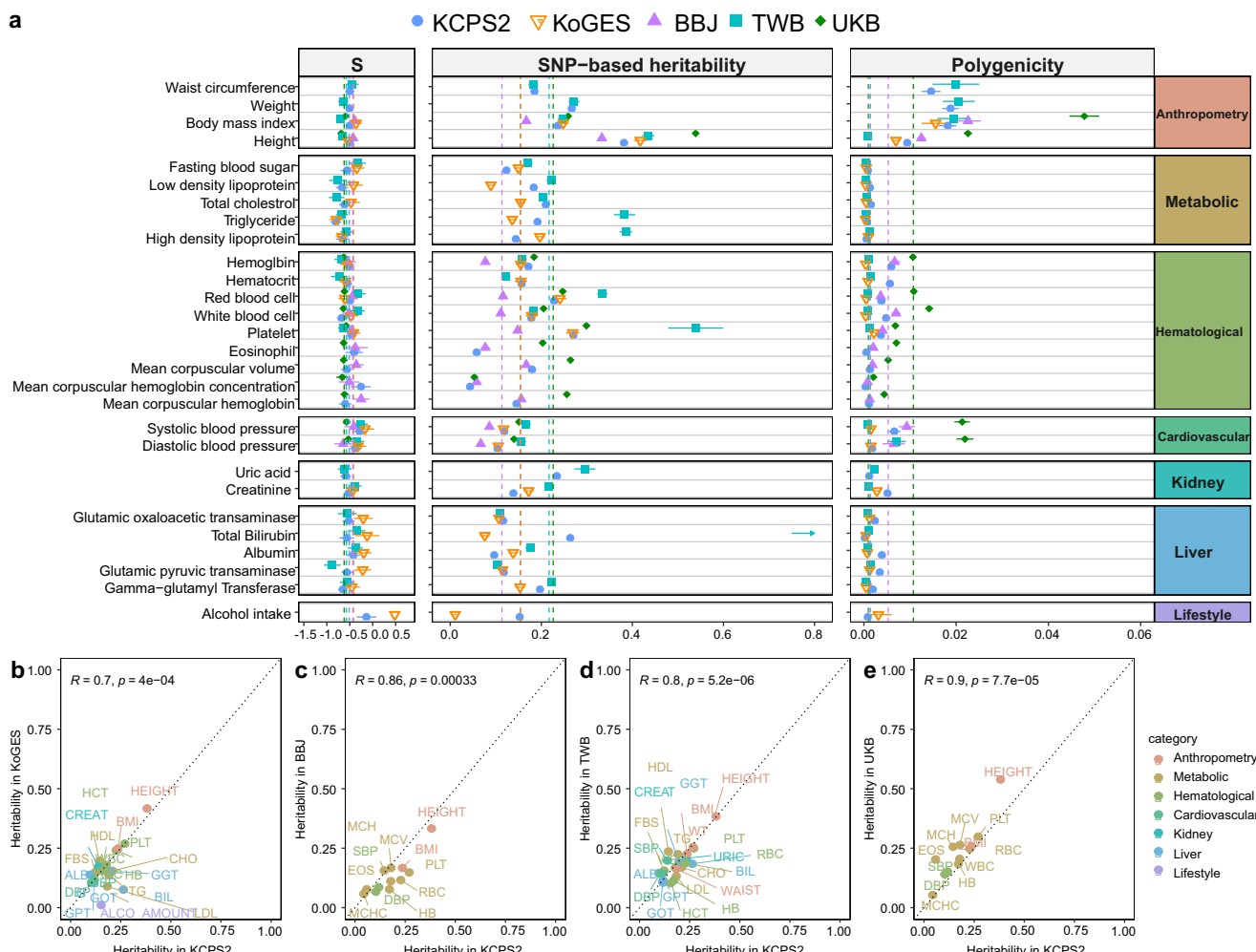

**Fig. 4 | Genetic architecture of complex traits across KCPS2 ($n = 153$ K), KoGES ($n = 72$ K), BBJ ($n = 179$ K), TWB ($n = 102$ K), and UKB ($n = 420$ K). a** The dots represent posterior means and horizontal bars represent standard errors of the parameters for each trait. The vertical dashed line shows the median of the estimates across traits. Full results are shown in Supplementary Data 8. Pearson correlations of SNP-heritability between KCPS2 and KoGES (**b**), BBJ (**c**), TWB (**d**), and UKB (**e**) across the traits shown in a, except for TWB. For heritability in TWB shown in (**d**), we used the heritability estimates reported by Chen et al.[15]. The comparisons between KCPS2 heritability estimates and TWB heritability estimates using SbayesS

and KCSP2 LD matrix are shown in Supplementary Fig. 6b. Data are presented as posterior means of SNP-heritability. The trait categories are indicated by different colors labeled with their trait names. For the eight traits available in all five studies, we observed high correlations of heritability between KCPS2 and the other biobanks: KoGES (Pearson correlation $r = 0.99$, 95% confidence interval [CI]: 0.97–1.00), BBJ ($r = 0.93$, 95% CI: 0.64–0.99), TWB ($r = 0.93$, 95% CI: 0.65–0.99), and UKB ($r = 0.97$, 95% CI: 0.82–0.99). All Pearson correlation tests were two-sided, and the reported $p$-values were not corrected for multiple testing.

---

discovery and provide insights into the biological mechanisms of human quantitative complex traits. We discovered 301 previously unreported loci that have lower median MAF in European ancestry individuals than in East Asian populations, which was enabled by leveraging samples from diverse ancestry groups. In particular, the novelty rate was high for TSH (32 out of 100 loci) in KCPS2. The most strongly associated lead variant with TSH, rs10799824-A (Beta = -0.14, $P = 2.98 \times 10^{-139}$), was previously reported in GWAS of TSH[35] (Beta = −0.11, $P = 4.0 \times 10^{-21}$) and strict autoimmune hypothyroidism[17] (Odds ratio=0.87, $P = 4.1 \times 10^{-18}$) in European ancestry individuals. Several lead TSH loci were mapped to nearby genes previously linked to thyroid function, including rs13030651-A (Beta = 0.035, $P = 1.09 \times 10^{-13}$) in the thyroid adenoma associated gene (THADA)[36,37] and rs2160215 (Beta=0.065, $P = 1.75 \times 10^{-51}$) in thyrotropin receptor gene (*TSHR*)[38,39]. Notably, a novel missense variant *CD36* p.Pro90Ser (rs75326924-T; Beta = −0.052, $P = 9.08 \times 10^{-11}$) found in our TSH GWAS (AF$_{KCPS2}$ = 0.068) has low frequency in the gnomAD v4.1.0 East Asian genetic ancestry group (MAF = 0.03) but is exceedingly rare outside of that group (MAF < $10^{-5}$) and entirely absent from European genetic

ancestry groups[40]. *CD36* (also known as fatty acid translocase [FAT]) facilitates the transport of fatty acids into cells and participates in triglyceride storage[41]. A study in hypothyroid rats showed a reduced fatty acid absorption in the liver compared to euthyroid rats[42] and decreased hepatic FAT expression has been demonstrated in rats with postnatal hypothyroidism[43]. Moreover, recent studies have revealed that *CD36* contributes to the tumorigenesis and development of multiple cancer types by reprogramming the metabolism of glucose and fatty acid[44–46], providing new insights for developing potential therapeutic target and prognostic biomarker in the clinical setting. We also found high novelty rate for GWAS of CEA in KCPS2, which recapitulates several known tumor biomarkers in various cancer types, including a non-coding transcript exon variant (rs149037075-T; Beta=0.179, $P = 4.8 \times 10^{-477}$) in *ABO*[47,48] and a missense variant (rs28362459-C; Beta = 0.068, $P = 4.85 \times 10^{-128}$) in *FUT3* (also known as Lewis gene)[49–52], consistent with previous CEA GWAS[53]. Previous studies show that determinants of the blood A and B antigens and Lewis antigens and of CEA share the same glycoprotein carrier molecules[54,55], which might explain the association of CEA concentrations with the *ABO* and the

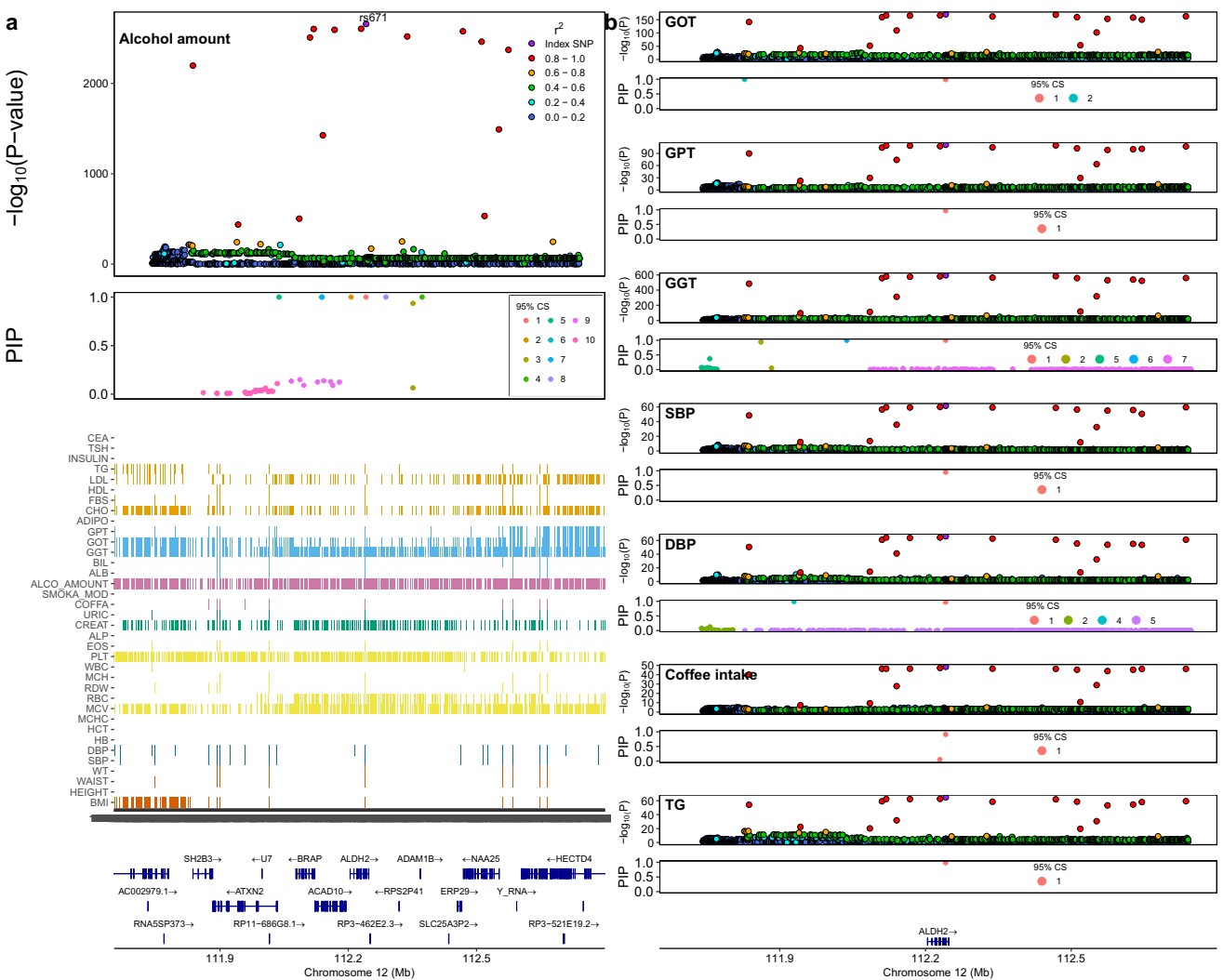

**Fig. 5 | Fine-mapping and colocalization analysis of ALDH2 region in KCPS2.**
**a** Association between ALDH2 (12q24.12) and alcohol intake in KCPS2, estimated using a linear mixed model. Colors in the Manhattan panels represent $r^2$ values to the lead variant rs671. In the posterior inclusion probability (PIP) panels, derived from SuSiE fine-mapping, only fine-mapped variants in the 95% credible sets (CS) are colored. The heatmap represents significant GWAS variants ($P < 5.0 \times 10^{-8}$) identified from linear mixed model analyses for the other quantitative traits.
**b** Colocalization analysis (performed using coloc) between alcohol intake and

seven traits that showed PIP > 0.9 for rs671 was done in the same region. All seven traits shown here had a posterior probability of colocalization at the specified region of 1 with alcohol intake. Each regional plot shows associations of each locus for the most significantly associated trait, which was all rs671 with PIP > 0.9. Full fine-mapping and colocalization analysis results are shown in Supplementary Data 10. All reported $-\log_{10}(P\text{-values})$ represent two-sided tests and uncorrected for multiple testing.

*FUT3* locus. Several variants not previously reported in CEA GWAS were mapped to genes with potential role in cancer such as *C15orf39*[56] (rs143001709; $P = 1.31 \times 10^{-8}$), *ST6GAL1*[57] (rs73187787; $P = 3.36 \times 10^{-11}$), and *CCDC138*[58] (rs10179849; $P = 6.55 \times 10^{-19}$). Further studies are warranted to investigate the potential functional importance of these associations.

As a global effort to broaden the population diversity of genetic studies in East Asia, the KCPS2 GWAS enhanced our understanding of the genetic basis of complex traits in a Korean population. Our genetic correlations across 36 quantitative traits recapitulated known biology, including negative genetic correlations of HDL, adiponectin, and bilirubin with cardiometabolic risk factors[25,27–29] and positive genetic correlations of smoking, alcohol intake, and hemoglobin with cardiometabolic traits[31,59]. Notably, most of the significant genetic correlations were consistent with phenotypic correlations, which underscores the robustness and potential of the genetics-based approaches to understand biological architectures of complex traits. Many of the significant findings were consistent with BBJ[8], KoGES[13], TWB[15], and

UKB[60], suggesting a similar genetic architecture for these quantitative traits within EAS populations and across EAS and EUR populations as shown by our work and previous findings on within- and cross-ancestry genetic correlation analysis[6,9,15].

Our findings provide opportunities to investigate the genetic architecture of complex traits within East Asian and across continental populations. While similar negative selection patterns were observed across traits and populations, the heritability estimates vary within EAS and across EAS and EUR populations, which may be attributed to several factors such as phenotype data collection, biobank design and environmental influences. For instance, compared to a hospital-based cohort such as BBJ, participants of a population-/community-based cohort such as KCPS2, KoGES, TWB, and UKB may have different distributions of disease-related traits due to healthy-volunteer effects[61]. Hence, the comparison of heritability estimates across biobanks requires careful consideration of technical differences, potential collider bias, and variability in baseline health status among studies[6]. Moreover, we demonstrated the correlation in heritability between

KCPS2 and TWB improved when the heritability of TWB were replaced by the previously reported estimates[15] that used LDSC and in-sample LD, especially for metabolic and hematological traits. Thus, in addition to the phenotype heterogeneity, heritability may be affected by different heritability estimation methods and LD matrices. Further research is needed to explore the impact of these factors on genetic architecture comparisons. Nevertheless, our study highlights the importance of increasing genetic diversity to understand genetic architecture of diverse populations, which is crucial to achieve equitable delivery of genomic knowledge to global populations[6,62,63].

The KCPS2 GWAS facilitated pinpointing causal variants through fine-mapping. For example, a missense variant rs671 ($AF_{EAS}$ = 0.2254 vs. $AF_{EUR}$ = $2.4 \times 10^{-5}$ in non-Finnish EUR populations in gnomAD v4.0.0[40]) was identified as the causal variant in the *ALDH2* gene for eight traits including alcohol intake, GGT, GOT, GPT, SBP, DBP, coffee intake, and triglyceride through fine-mapping. *ALDH2* gene is the target of drug for alcoholism which irreversibly inactivate catalytic Cys302 in *ALDH2* by carbamylation in the substrate site of the enzyme[64,65]. Furthermore, our colocalization results suggest that alcohol intake is a causal risk factor for liver enzymes (GGT, GOT, GPT), blood pressure, and triglyceride levels at the rs671 region, which is supported by evidence for causation from previous MR studies among East Asian populations[30,66,67]. An additional pleiotropic effect of rs671 across dietary habits on foods and beverages (coffee, green tea, yogurt, natto, tofu and fish) was previously reported, with the same directional effects of rs671 on coffee consumption (rs671-A, Beta = 0.061, *P* = 3.4e-49) in our study[68]. Our findings demonstrated the potential of diversifying EAS GWAS to uncover genetic associations that are common in EAS populations but rare in EUR populations, which could not be discovered even with very large European sample sizes. Furthermore, discovery of such variants may help identify targets for prevention and treatment, thus offering equitable access to precision medicine to diverse populations.

We note several limitations to our study. First, we only conducted GWAS of continuous traits due to limited power for disease phenotypes. Further investigation into disease outcomes should be conducted. Second, we conducted fine-mapping in KCPS2 only for a particular locus, which might cause a concern about potential LD tagging effects for observed pleiotropy. Recent studies suggest that multi-ancestry fine-mapping can improve refinement of causal variants by leveraging different LD patterns across ancestries[69,70]. We will explore these potential extensions in the near future. Third, for the estimation of genetic architecture parameters, in-sample LD was used for KCPS2, BBJ, and UKB but not for TWB. Since we were unable to find publicly available data to estimate LD in a Taiwanese population, we estimated the genetic architecture parameters using the LD matrix based on the 50 K individuals from KCPS2. Such disagreement between the genetic associations and the correlation matrix may induce spurious results due to different LD patterns between the Taiwanese population and Korean individuals from KCPS2, even though both are East Asian populations.

Our findings highlight how broadening the population diversity of GWAS samples can aid discovery and post-GWAS analyses. Our results also provide insights into the genetic architecture of complex traits in East Asian populations. By increasing the sample size and ancestral diversity of GWAS samples, our analysis may help identify novel population-specific targets for prevention and treatment, thus offering equitable access to precision medicine to diverse populations.

## Methods
### Study population
The Korean Cancer Prevention Study-II Biobank (KCPS2) is a prospective cohort study based in Korea with genotype data and measurements of a wide range of phenotypes collected from 153,950 subjects (Fig. 1). Participants in KCPS2 undertook routine health assessments at nationwide health promotion centers between 2004 and 2013. Approximately 90% of participants were recruited from the Seoul and Gyeonggi regions, where about 40% of the South Korean population resides (around 19 million people). Mean age of participants at recruitment was 41.7 yr old, and 40% were female. The study design and recruitment have been described in detail previously[22]. KCPS2 collects extensive phenotypes including demographics, socioeconomic status, environmental exposures, lifestyle, dietary habits, family history and self-reported disease status through structured questionnaires. The anthropometric measures as well as blood and urine samples were collected at recruitment, and several biomarkers were assayed subsequently. All participants in the KCPS2 were genotyped using either the Illumina Global Screening Array (GSA) v2.0 (78,260 samples) or the Korean Chip array v1 (90,245 samples). All participants provided written informed consent before participation.

Quality control (QC) and imputation were conducted separately for each of the two SNP arrays. First, SNPs with low call rate (<95%) were filtered out, along with samples with low call rate (<98%), gender discrepancy, excessive heterozygosity, excessive singletons, and duplicates. Additionally, SNPs with Hardy-Weinberg equilibrium *p*-value < $10^{-4}$ or minor allele frequencies (MAF) < 0.01 were excluded. Following QC, the data were phased using SHAPEIT4[71] and imputed using IMPUTE5[72] with 1000 Genomes Project Phase 3 data. Variants with imputation INFO < 0.8 were excluded after imputation. The two imputed data of GSA chip and the Korean chip were then merged, resulting in a total of 6,809,738 overlapping variants.

### Korean Genome and Epidemiology Study (KoGES)
We used KoGES GWAS summary statistics from Nam et al. (https://zenodo.org/record/7042518)[13]. KoGES is a population-based prospective cohort study comprising 72,000 Korean men and women who are recruited from the national health examinee registry at baseline. Mean age of participants at recruitment was 54.15 yr old, and 64% were female. Genome-wide association tests were performed for 76 phenotypes, using linear or proportional odds logistic mixed models implemented in SAIGE (v0.44.5)[23] or POLMM[73] adjusted for top 10 principal components (PC), age, sex, and adjustment of assessment details such as cohort and year of examination. We extracted 22 quantitative traits from KoGES that matched the quantitative traits we analyzed in KCPS2: albumin (ALB), alcohol amount (ALCO_AMOUNT), gamma-glutamyl Transferase (GGT), glutamic oxaloacetic transaminase (GOT), body mass index (BMI), creatinine (CREAT), diastolic blood pressure (DBP), glutamic pyruvic transaminase, (GPT), fasting blood sugar (FBS), hemoglobin (HB), high density lipoprotein (HDL), height (HEIGHT), hematocrit (HCT), low density lipoprotein (LDL), platelet (PLT), red blood cell (RBC), systolic blood pressure (SBP), total bilirubin (BIL), total cholesterol (CHO), triglyceride (TG), waist circumference (WC), and white blood cell (WBC). The quantitative traits used in the KoGES GWAS were inverse rank-based normal transformed.

### Biobank Japan (BBJ)
We used GWAS summary statistics from BBJ from Sakaue et al. (https://pheweb.jp/)[9]. BBJ is a hospital-based cohort, consisting of 179,000 Japanese participants. The mean age of participants at recruitment was 63.0 years, and 46.3% were female. Genome-wide association tests were conducted for 220 phenotypes, using linear or logistic mixed models implemented in SAIGE (v0.37)[23] or BOLT-LMM (v.2.3.4)[74]. The measured values of each quantitative trait were adjusted for age, $age^2$, sex, age by sex interaction, $age^2$ by sex interaction, and the top 20 PCs, and the residuals were then transformed by rank-based inverse normalization. We extracted 27 quantitative traits from BBJ that matched the quantitative traits we analyzed in KCPS2: LDL, TG, HDL, CHO, FBS, GOT, GPT, GGT, BIL, ALB, CREAT, uric acid (URIC), alkaline phosphatase (ALP), HB, HCT, mean corpuscular hemoglobin (MCH), mean corpuscular hemoglobin concentration (MCHC), mean corpuscular

volume (MCV), RBC, WBC, PLT, eosinophil (EOS), SBP, DBP, weight (WT), HEIGHT, and BMI.

### Taiwan Biobank (TWB)

TWB is a population-based prospective cohort study of the Taiwanese population, comprising 102,900 participants. The mean age of participants at recruitment was 50.0 years, and 64% were female. We used two sets of TWB GWAS summary statistics from Chen et al. [15] for the meta-analysis and the genetic architecture analysis, respectively. For the meta-analysis, we used the TWB GWAS summary statistics, using linear mixed models implemented in Regenie (v1.0.5.4)[75] adjusted for age, age[2], sex, age by sex interaction, age[2] by sex interaction, and top 20 PCs. We extracted 23 quantitative traits from BBJ that matched the quantitative traits we analyzed in KCPS2: LDL, TG, HDL, CHO, FBS, GOT, GPT, GGT, BIL, ALB, CREAT, URIC, HB, HCT, RBC, WBC, PLT, SBP, DBP, WT, HEIGHT, BMI, and WAIST. All phenotypes used in the TWB GWAS were inverse rank-based normal transformed. For the genetic architecture analysis, we used TWB GWAS summary statistics using linear regression implemented in PLINK2[76], as the previously reported genetic architecture estimates we compared against were also based on linear association test statistics.

### UK Biobank (UKB)

The UKB project is a population-based prospective cohort that recruited ~500,000 participants from across the United Kingdom. The mean age of participants at recruitment was 56.8 years, with 53.8% identifying as female. All GWAS summary statistics from UKB used in this study were publicly available at: https://pan.ukbb.broadinstitute.org/, generated and released by Karczewski et al[16]. For our analysis, we used the EUR ancestry estimates from the Pan-UKB data. We included 29 GWAS from UKBB: Height, BMI, HB, HCT, RBC, WBC, PLT, SBP, DBP, WT, LDL, TG, HDL, CHO, FBS, GGT, BIL, ALB, CREAT, ALCO_AMOUNT, GOT, GPT, URIC, ALP, MCH, MCHC, MCV, EOS, and WAIST. All phenotypes used in these GWAS were inverse rank-based normal transformed.

### Genome-wide association analysis in KCPS2

We performed GWAS on 36 quantitative traits including anthropometric measures and biomarkers spanning 8 categories (metabolic, liver, thyroid hormone, tumor marker, kidney, hematological, cardiovascular, arthrometry, and lifestyle factors). For each trait, we excluded samples with measurements that were more than six standard deviations away from the sample average.

We used linear mixed models implemented in SAIGE (v.1.1.9)[23] for association testing, controlling for age, sex, 10 PCs, and SNP array. The SAIGE method contains two main steps: in step 1, we used a subset of linkage disequilibrium (LD)-pruned variants with $R^2 < 0.2$ (158,729 variants) to obtain the genetic relationship matrix. We included age, sex, 10 PCs, and SNP array as covariates in step 1. Single-variant association testing was performed in step 2 where the phenotypes were inverse rank-based normal transformed and leave-one-chromosome-out scheme to remove the proximal contamination. We used FUMA[77] with 1000 Genome Project Phase 3[78] EAS samples as LD reference to identify independent genome-wide significant loci ($p < 5 \times 10^{-8}$) for each trait, window size of 500 kb, and LD threshold $R^2$ of 0.1.

Linkage disequilibrium score regression (LDSC)[79] was applied to estimate cross-trait genetic correlations ($r_g$) in KCPS2. We ran stratified-LDSC (S-LDSC)[80] with a full baseline-LD v1.2 model[80] to compute LDSC intercept. To correctly specify effective sample size in LDSC or S-LDSC analysis, we used GWAS summary statistics generated from simple linear regression models instead of linear mixed models, which have a different effective GWAS sample size than the study sample size[81]. We ran linear regression using PLINK2[76] for association testing, controlling for age, sex, 10 PCs, and SNP array in unrelated KCPS2 samples. Specifically, we removed second degree or more

closely related individuals using the software KING[82]. All phenotypes used in these GWAS were inverse rank-based normal transformed.

### Novel association identification

We mapped each trait to a term in the Experimental Factor Ontology (EFO) each trait (Supplementary Data 3). For each of the independent loci we identified to be associated with a given trait, we queried the Open Target Genetics database (release 22.09)[83,84] for each of the independent loci we identified to be associated with a given trait to identify any previously reported associations (with the same EFO term or category, see below). We considered a trait-associated locus as novel ("previously unreported") when the locus with ±500 kb flanking window did not include any lead variants that were previously reported with the same EFO term. Given the widespread pleiotropy and phenotypic heterogeneity[85], we may overcount novel associations. We therefore also used EFO categories, which are more generic than EFO terms to evaluate novelty. For example, "body height" (EFO_0004339) is an EFO term that maps to the broader EFO category of "body measurement" (EFO_0004324) by GWAS Catalog[24]. We further exhaustively searched for previous reports of genetic association in a given trait using the GWAS Catalog, which might not be included in the Open Target Genetics database. Since the recent GWAS results of height[86] and TSH[87,88] were not listed in the GWAS Catalog at the time of the curation, we additionally excluded variants that were genome-wide significant in the GWAS. Since the results by KoGES[13] and TWB[15] were not listed in the Open Target Genetics database or the GWAS Catalog at the time of evaluation for the novel association, we additionally excluded loci with ±500 kb flanking window that include any previous associations reported by KoGES or TWB. A flow chart illustrating the process for identifying novel loci is demonstrated in Supplementary Fig. 8.

### Evaluation of gene pleiotropy

We investigated gene pleiotropy, where a gene affects multiple traits in KCPS2. We defined the degree of pleiotropy as the number of significant associations per gene ($p < 5 \times 10^{-8}$). The list of genes mapped to each SNP in KCPS2 GWAS results was taken from FUMA[77] to map SNPs in GWAS results to a gene with the 1000 Genome Phase 3[78] EAS reference panel. We then quantified the degree of pleiotropy per gene by aggregating and counting the number of genome-wide significant associations across 36 traits.

### Meta-analysis of EAS and EUR GWAS

We conducted meta-analysis of 21 traits across KCPS2, KoGES, BBJ, TWB, and UKB (European ancestry samples) to further identify novel loci across East Asian and European ancestry populations. We implemented inverse-variance-weighted fixed-effect meta-analysis using METAL[89]. We retained variants presented in at least one of the studies in this meta-analysis for loci discovery in the EAS and EUR populations. We then used FUMA[77] to identify genome-wide significant loci in the meta-analysis after clumping variants with $p$-values $< 5 \times 10^{-8}$, window size of 5 Mb, and LD threshold $R^2$ of 0.1. We identified the association as novel if none of the variants within the locus reached genome-wide significance ($P < 5 \times 10^{-8}$) in KoGES, BBJ, TWB, or UKB GWAS.

### Genetic architecture of complex traits in KCPS2, KoGES, BBJ, TWB, and UKB

We used SbayesS[90] to estimate the SNP-based heritability ($h^2_g$), polygenicity (π; proportion of SNPs with nonzero effects), and the relationship between minor allele frequency (MAF) and SNP effects ($S$ parameter) for 36 traits in KCPS2. We constructed a full LD correlation matrix based on 50 K unrelated individuals from KCPS2 and shrunk the matrix to ignore small LD correlations due to sampling variance using the shrinkage method from Wen and Stephens[91]. To calculate the LD matrix shrinkage estimate, we used a genetic map for East Asian

populations, with the effective population sample size of 12,239[78], while using the default shrinkage cutoff ($10^{-5}$). We then compared the genetic architecture of KCPS2 with KoGES, BBJ, TWB, and UKB across six categories including anthropometry, cardiovascular, hematological, kidney, liver, and metabolic traits (among which 22 traits overlap between KCPS2 and KoGES, 12 traits overlap between KCPS2 and BBJ/UKB, 23 traits overlap between KCPS2 and TWB, and 8 traits available in all biobanks: height, body mass index, platelet, red blood cell, white blood cell, hemoglobin, systolic blood pressure, and diastolic blood pressure). When comparing the median SNP-heritability of trait categories between two studies, we restricted the comparison to the same list of traits within each category for that specific biobank pair and performed a Wilcoxon signed-rank test to assess statistical significance. For BBJ and UKB, we used the previously reported genetic architecture parameter estimates[62], which were constructed using GWAS summary statistics generated from linear regression models and in-sample LD for the corresponding unrelated population. For a fair comparison of these parameters between KCPS2, BBJ, TWB, and UKB, we applied SbayesS to GWAS summary statistics generated from linear regression models in unrelated KCPS2 and TWB, instead of linear mixed models. Since summary statistics from linear regression models were not publicly available for KoGES, we used linear mixed models instead. For TWB, we estimated the genetic architecture parameters using the LD matrix based on the 50 K individuals from KCPS2 because we were unable to find publicly available data to estimate LD in a Taiwanese population. As we noted in our limitation, presumably because unmatched LD was used for TWB, we found overestimation of SbayesS heritability estimation in TWB. This overestimation was magnified in total bilirubin on chromosome 2 ($h^2_g = 1.12$), which is a Mendelian locus, harboring the *UGT1A1* gene. Thus, we removed chromosome 2 when estimating correlations between heritability in KCPS2 and TWB (Supplementary Fig. 6b).

## Cross-biobank genetic correlation
To estimate cross-biobank genetic effect correlations within EAS (KCPS2-KoGES, KCPS2-BBJ, and KCPS2-TWB), we used LDSC[79] to estimate rg using the 1000 Genomes phase 3 EAS reference panel. We used Popcorn (v.1.0)[92] to estimate cross-biobank genetic-effect correlation between KCPS2 and UKB GWAS with precomputed cross-population scores for EUR and EAS 1000 Genomes Project populations provided by the authors. For a fair comparison, we restricted to Hap-Map3 SNPs that were shared across all five biobanks. We applied the analysis to traits with heritability calculated by LDSC or Popcorn >0.01 and their GWAS summary statistics generated from linear mixed models from all biobanks which were publicly available.

## Fine-mapping and colocalization analysis
We fine-mapped one of the most pleiotropic regions identified by GWAS of the 36 traits above, a 500 kb region flanking *ALDH2* in KCPS2. We applied SuSiE[33] to GWAS summary statistics and in-sample LD on 1476 SNPs in this region. We implemented colocalization analysis to further investigate whether two traits share a causal variant. We applied coloc.susie[93] which allows multiple signals to be distinguished using SuSiE, and then performed colocalization analysis on all possible pairs of signals between the traits. We performed colocalization analysis in a 500 kb window centered on an identified causal variant between alcohol intake and the other traits with PIP of rs671 being greater than 0.9 from fine-mapping results. We reported posterior probability of colocalization (PP4) for each of these pairs at the specified region. We applied LocusZoom[94] to visualize the colocalization analysis.

## Reporting summary
Further information on research design is available in the Nature Portfolio Reporting Summary linked to this article.

## Data availability
The GWAS summary statistics generated in this study are publicly available at https://zenodo.org/records/15132424. Data sources for other publicly available GWAS summary statistics are available in Supplementary Data 13. The summary statistics for KoGES used in this study were downloaded from the KoGES Zenodo (https://zenodo.org/record/7042518), BBJ summary statistics from the Biobank Japan PheWeb (https://pheweb.jp/), TWB summary statistics from GWAS Catalog (https://www.ebi.ac.uk/gwas/publications/38116116), and summary statistics for Europeans in UKB were downloaded from Pan-UK Biobank (https://pan.ukbb.broadinstitute.org/).

## Code availability
We used publicly available software for the analyses. The software used is listed and described in the Methods section of our manuscript. Analysis code used in this manuscript is publicly available at https://doi.org/10.5281/zenodo.15110489.

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

## Acknowledgements

We acknowledge the participants of KCPS2 and the management team and leadership of KCPS2 for their outstanding support in collecting samples and clinical data. We thank KoGES, BBJ, TWB, and UKB for providing resources and releasing the GWAS summary statistics, which made this study possible. K.J.J. acknowledged support from the Basic Science Research Program through the National Research Foundation of Korea funded by the Ministry of Education (RS-2023-00239122). R.D. was supported by R01 GM148494. A.L.P. was supported by R01 HG006399. A.R.M. was supported by funding from the National Institutes of Health (K99/R00MH117229) and funding from a Broad Next Generation Fund. P.K. was supported by U01CA261339.

## Author contributions

Y.H.J., P.K., and A.R.M. wrote the manuscript. Y.H.J. performed data analysis. Y.W. provided analysis guidance. K.J.J., J.Y.L., and H.K. contributed to data collection. P.K., A.R.M., A.L.P., and R.D. supervised the study. All authors reviewed and contributed to the manuscript.

## Funding

## Competing interests

The authors declare no competing interests.

## Additional information

**Supplementary information** The online version contains
supplementary material available at

Keum Ji Jung or Peter Kraft.

**Peer review information** *Nature Communications* thanks Seunggeun
Lee and the other, anonymous, reviewer(s) for their contribution to the
peer review of this work. A peer review file is available.

