## [Transparent Peer Review file · Nature Communications]

Genome-wide association studies in a large Korean cohort identify quantitative trait loci for 36 traits and illuminate their genetic architectures

Corresponding Author: Dr Peter Kraft

Version 0:

Reviewer comments:

Reviewer #1

(Remarks to the Author)

This paper presents GWAS conducted using data from the Korean Biobank, KCPS2, which is an welcome addition to the GWAS resources for diverse ancestry groups. The authors compared the GWAS results with those from existing East Asian biobank studies and the UK Biobank, followed by meta-analysis and fine-mapping. The manuscript is well-written, and I have only a few relatively minor comments

1. It is interesting that the regression slope of effect sizes from KCSP2 and those from BBJ is nearly one. Initially, I assumed this indicated perfect concordance, but upon examining the plot, I noticed variations. Did you constrain the regression line to pass through the origin? If so, the slope is not equivalent to the correlation coefficient. It would be beneficial to also report the correlation coefficient and the mean squared error.

2. Figure 5. The Susie analysis shows the potential causal variants beyond rs671, but I am wondering whether this is due to the high LD with rs671. If a more traditional conditional analysis was performed with rs671 as a covariate, would all association signals in the region (as shown in the heatmap) be eliminated? The authors conducted fine-mapping with $L=1$, but a more straightforward approach could visually demonstrate whether the associations in this region are entirely driven by rs671 or not.

3. Novel Loci: The GWAS of KCPS2 identified a large number of novel loci (616). However, I wonder if this high number is due to the comparison being limited to loci reported by Open Targets. If significant loci from other East Asian biobanks were also considered as known, how many of these variants would still be classified as novel? Additionally, among these novel loci, how many can be replicated in other East Asian biobanks?

minor typos

Ex. Line 520 : aAnthropometric

Reviewer #2

(Remarks to the Author)

This paper reported a quantitative trait GWAS study in the Korean Cancer Prevention Study-II (KCPS2) Biobank. This biobank contains 153,950 participants, and the study focused on 36 quantitative traits. They reported 616 novel genetic loci in KCPS2 and meta-analyzed them with Biobank Japan and Taiwan Biobank, etc. Overall, these results contribute to the study of the genetic architecture of complex traits and diseases within East Asia, which is currently scarce.

I have the following comments:

1. KCPS2 cohort characteristics need to be presented and commented. Ideally, it needs to be compared with other Korean and/or East Asian cohorts.
2. How does this biobank compare with other Korean biobanks mentioned in the paper, e.g., KoGES? Does the sample overlap with each other?
3. I am also curious about the coverage of the overall Korean population. Are they mostly from big cities or rural areas, south or north? The South Korean population is 51 million, and KCPS2 covers only 0.2%. Are they enriched with people having cancer risk factors?
4. How were the biomarkers QCed? Are they from the EHR? Were these measured once or multiple times?
5. Heritability was estimated to be lower than usual. Is it due to the use of summary statistics from GWAS? What if individual data is used to estimate heritability?
6. The novel rate of thyroid-stimulating hormones is high. Any reason for this? Is there any gender difference for this, given thyroid dysfunction is more common in females?
7. How to interpret $r_g > 0$, but r_p is zero or very close to zero?
8. Why was the correlation with KCPS2 effect sizes greatest for the BBJ and not the other Korean cohort? It would be really helpful to compare the cohort characteristics...
9. Why meta-analyze with UKB? Together with all the East Asian cohorts?

Reviewer #3

(Remarks to the Author)

In this manuscript, the authors performed GWAS on 36 quantitative traits in KCPS2, a large Korean Biobank with 153,950 individuals. They identified 616 novel genetic loci in KCPS2 and 3,524 loci in a meta-analysis with the Korean Genome and Epidemiology Study (KoGES), Biobank Japan (BBJ), Taiwan Biobank (TWB), and the UK Biobank (UKB). They described differences in the genetic architecture between East Asian and European populations, and highlighted East Asian specific associations. The manuscript is well structured and clearly written, and I greatly appreciate the effort in improving the population diversity of GWAS. I have a few questions and suggestions for the authors, which I listed below.

1. I think a major limitation is that the genetic architecture analysis was not performed in KoGES, given that it was the only other large Korean study included in this manuscript. The authors suggested that summary statistics from linear regression models were not available, but it was unclear how sensitive their analyses would be using summary statistics from linear mixed models instead (i.e., how did these summary statistics compare with each other). For example, they estimated cross-trait genetic correlations in KCPS2 using LDSC and summary statistics from linear regression models, but their cross-biobank genetic correlations between KCPS2 and KoGES were likely estimated using LDSC and summary statistics from linear mixed models (at least in KoGES). Many GWAS summary statistics-based methods were originally derived in the linear regression framework, but they have been applied to GWAS summary statistics from more complex models such as linear mixed models. On the other hand, the sample size in KoGES (72K) was not dramatically smaller than TWB (102K). Some further justifications for leaving out KoGES in this analysis would be helpful.
2. The validity of simple linear regression needs to be justified, as it could lead to spurious association findings in the presence of relatedness. For UKB, GWAS summary statistics from linear regressions (lines 592-594 on page 29) are likely invalid, given that the sample size (420K) suggested that related individuals had been included.
3. The novel loci in KCPS2 tend to be more common than in 1000G EUR samples (lines 90-92 on page 5). What about the MAF of these novel loci in other East Asian samples (e.g., 1000G EAS, or BBJ, KoGES, TWB)?
4. How were the lead variants defined in Figure 3? For one locus, it is possible that the lead variant in KCPS2 might be different from the lead variant other biobanks (or the lead variant from meta-analysis). If the lead variants were different across studies, were they in LD with each other? How about their MAF's? Also I find it a little difficult to interpret the results that the correlation between KCPS2 and BBJ effect size estimates was greater than the correlation between KCPS2 and KoGES effect size estimates (lines 144-145 on page 8), given that both KCPS2 and KoGES samples were Koreans. Was that caused by the different traits analyzed across these studies, and/or different lead variants across studies?
5. I am not sure if median SNP-heritability would be a good comparison between KCPS2 and BBJ since the traits analyzed were different across studies (line 171 on page 10). Some traits might have a higher/lower heritability but were unavailable in some studies. Was the difference between 0.26 and 0.25 statistically meaningful? The benchmark could be more rigorous if performed on the same traits across all studies. Also I am not sure if correlation of heritability estimates (lines 175-178 on page 10) is the best measure, given limited numbers of traits (and different traits analyzed across studies). Heritability estimates could be highly correlated, but still quite different between two studies (e.g., 0.6, 0.4, 0.2 for three traits in one study, and 0.3, 0.2, 0.1 in another study). Was the reported lower correlation with TWB (line 176 on page 10) statistically significant? The confidence intervals were very wide (possibly due to the limited numbers of traits) and they appeared to overlap with each other.
6. I agree with the authors that using in-sample LD is critical, since LD scores could be different in KCPS2 and TWB. It might be better to move TWB in-sample LD results to Figure 4, and TWB results using LD scores based on 50K KCPS2 samples to Supplementary Figures.
7. The authors mentioned that phenotypes were rank normal transformed (line 542 on page 27). Does that mean all the reported beta's were effect size estimates on the transformed scale (i.e., not on the raw phenotypes)? Could the authors

confirm that the same rank normal transformation was performed for all these phenotypes in the KoGES, BBJ, TWB and UKB GWAS summary statistics? Otherwise the inverse-variance-weighted fixed-effect meta-analysis should not be performed if the effect size estimates were on different scales.

Minor:

1. I am not sure if the direction from “meta-analysis” to GWAS is accurate in Figure 1. I thought GWAS (in KCPS2) should precede meta-analysis?
2. It would be great to improve the alignment in Figure 2a. Currently there is a slight shift due to the extra space between trait groups on the left panel (but not on the right panel), so they do not align well horizontally. The font size was too small for “Thyroid hormone” and “Tumor marker” on the left panel.
3. After revising Figure 2a, it would be great to also format Figure S3 in the same style.
4. Gene names in Figure 2b could be better presented if they are annotated horizontally near the dots. Vertical lines look a little confusing when they stack together.
5. In Figure 2c, the INSULIN-CEA pair was shown in blue (only r_p significant) but the point estimate was very close to 0 for r_p and largely negative for r_g . It would be nice to double check. In addition, if the slope between pairwise r_g and r_p was estimated from a linear regression model, I think the standard error estimate (line 116 on page 7) might not be very informative, since the independence, normality and homoscedasticity assumptions were likely violated for r_g . Also, which FDR threshold was used?
6. Since the coding allele in GWAS might be arbitrary, it would be better to present Figure 3b and Figure S4 as plots versus the MAF (instead of AF) on the x axis.
7. Visualization could be improved in Figure 4a. The current overlay and alignment make it difficult to tell which dots came from which traits.
8. It is unclear whether $\text{neglog}P$ in Figure 5 refers to natural log or log base 10. For p-values, I think log base 10 would be more informative. In Figure 5b, trait labels were truncated at the top. If coloc.pp4 was all equal to 1 for these 7 traits, it could be removed from this figure and described only in the text (e.g., “posterior probability of colocalization at the specified region”, instead of “PP4”).
9. In the Discussion, when presenting beta or odds ratio for a variant, it would be great to specify which allele the authors were referring to. Flipping the alleles could change the direction of effect.
10. Typos: “wWe” in line 556 on page 27; “Zendo” in line 631 on page 31.

Reviewer #4

(Remarks to the Author)

Version 1:

Reviewer comments:

Reviewer #1

(Remarks to the Author)

The authors addressed all of my comments. I don't have any additional ones.

(Remarks on code availability)

I checked the GitHub page, and it appears that all the relevant codes are available.

Reviewer #3

(Remarks to the Author)

I appreciate the authors' efforts in addressing my previous comments. The revised manuscript has been improved in its clarity. I only have two minor suggestions for them to consider in Figure 5b:

1. The authors have removed coloc.PP4 in the figure since it was equal to 1 for all seven traits. I would suggest replacing “Coloc.PP4 represents the posterior probability of colocalization at the specified region. All seven traits shown here had $\text{coloc.PP4}=1$ with alcohol intake” by something like “All seven traits shown here had a posterior probability of colocalization at the specified region of 1 with alcohol intake.”
2. I would suggest increasing the width of the 95% CS legend boxes so that they don't have a second row to overlay with the lines of dots near $\text{PIP} = 0.0$ for GGT and DBP.

(Remarks on code availability)

Reviewer #4

(Remarks to the Author)

(Remarks on code availability)

SUBMITTED MANUSCRIPT (A VERSION TO RESUBMIT):

REVIEWER COMMENTS

We thank the reviewers for their careful and thoughtful reviews. In response, we have performed additional analyses (e.g. stepwise conditional analyses assessing independent signals at the *ALDH2* locus and meta-analyses of East Asian biobanks replicating novel loci identified in KCPS2), provided additional information (e.g. details on design of the KCPS2, KoGES, BBJ, TWB and UKB), and corrected inadvertent textual errors. In addition, due to the unique format of TWB summary statistics for several traits, previous meta-analysis did not account for these traits. We have rerun meta-analysis and corrected the number of genome-wide significant loci for meta-analysis accordingly. The reviewers' comments have helped us improve the manuscript.

Reviewer #1 (Remarks to the Author):

This paper presents GWAS conducted using data from the Korean Biobank, KCPS2, which is an welcome addition to the GWAS resources for diverse ancestry groups. The authors compared the GWAS results with those from existing East Asian biobank studies and the UK Biobank, followed by meta-analysis and fine-mapping. The manuscript is well-written, and I have only a few relatively minor comments

1. It is interesting that the regression slope of effect sizes from KCSP2 and those from BBJ is nearly one. Initially, I assumed this indicated perfect concordance, but upon examining the plot, I noticed variations. Did you constrain the regression line to pass through the origin? If so, the slope is not equivalent to the correlation coefficient. It would be beneficial to also report the correlation coefficient and the mean squared error.

Thank you for this observation. In our analysis, we used the `geom_smooth()` function in R with the formula `'y ~ x'`, which fits a standard linear regression model without constraining the line to pass through the origin. This model includes both an intercept and a slope. All the fitted intercepts were quite close to 0 ($< 1e-3$).

As you note, the slope from this regression is not equivalent to the correlation in effect estimates across biobanks. The slope and correlation will only be identical if the sample standard deviations of the 'X' and 'Y' variables are equivalent. We now present the correlation between the estimates from KCPS2 and the other biobanks in the text (page 10). We present the MSE in the table below, but have not added the MSE to the manuscript, because the qualitative

comparisons across biobanks based on the correlation and on the MSE are similar (KoGES, BBJ and TWB estimates have the same correlation with KCPS2 estimates and predict KCPS2 estimates with the same MSE), and because the correlation coefficients are easier to interpret in this case (we are measuring similarity between the two sets of estimates, not predicting one from the other).

Study	Regression coefficient (SE)	Correlation coefficient (MSE)
KoGES	0.871 (0.0048)	0.89 (1e-04)
BBJ	1.065 (0.0064)	0.89 (1e-04)
TWB	0.874 (0.0047)	0.89 (1e-04)
UKB	0.573 (0.0051)	0.765 (2e-04)

We have now replaced the regression coefficients by the correlation coefficients in the main text and this changes the qualitative assessment of relative agreement in effect sizes across biobanks:

The correlations between KCPS2 effect sizes and those in other East Asian biobanks are similar (KCPS2-KoGES: 0.890, KCPS2-BBJ: 0.887, KCPS2-TWB: 0.893), with smaller correlation with UKB (KCPS2-UKB: 0.765).

2. Figure 5. The Susie analysis shows the potential causal variants beyond rs671, but I am wondering whether this is due to the high LD with rs671. If a more traditional conditional analysis was performed with rs671 as a covariate, would all association signals in the region (as shown in the heatmap) be eliminated? The authors conducted fine-mapping with L=1, but a more straightforward approach could visually demonstrate whether the associations in this region are entirely driven by rs671 or not.

To address the concern about potential high LD with rs671, we performed additional analyses using **GCTA COJO**, a conditional analysis with rs671 as a covariate (columns L–N in Supplementary Table 10), as well as a stepwise selection analysis with two p-value cutoffs: Bonferroni-corrected ($P = 0.05 / \text{the number of fine-mapped variants}$) and genome-wide significance ($P = 5e-8$) (columns O–T). The detailed results are presented in Supplementary Table 10.

The overall patterns of independent signals identified through these two additional analyses were consistent with each other and the original SuSiE analysis. For traits such as **TG**, **GPT**, **SBP**, and **DBP**, where the fine-mapping credible set had only rs671 with a PIP > 0.95, most of the significance disappeared after conditioning on rs671. In these cases, rs671 remained the only independent signal following stepwise selection (under both p_bonf and p_gw thresholds).

For **alcohol intake**, conditioning on rs671 led to a substantial reduction in significance for other variants in the **ALDH2** region (from min $P = 5e-2000$ to min $P = 5e-40$) and a decrease in the

number of genome-wide significant variants in the **ALDH2** region (from 1094 to 494). However, some significant effects persisted even after adjusting for rs671. For instance, **rs11066079** (conditional P-value = 1.47e-46) emerged as the second most significant variant, with $r^2=0.0127$ relative to rs671, as shown in the figure below. Notably, among the seven credible sets with a size of 1 and a PIP = 1, three variants (rs671, rs11066079, and rs61055528) were either found in, or in LD ($r^2>0.6$) with, the independent signals identified by stepwise selection.

The figure shows associations between ALDH2 (12q24.12) and alcohol intake in KCPS2, after conditioning on rs671. Colors represent r^2 values to the lead variant rs11066079. Following rs11066079, the next significant SNP was rs11066080 (conditional P-value = 2.24e-46).

In conclusion, our conditional analyses confirm that the fine-mapping results are detecting independent signals, and the associations of alcohol intake in the region are not entirely driven by rs671.

We now included the conditional analysis results in the main text and Supplementary Table 10.

To assess the sensitivity of these results to analytic approach, we used conditional and joint analysis (COJO) to conduct association analyses in the ALDH2 region conditional on rs671 and perform a stepwise selection analysis. These approaches identified independent signals consistent with each other and the original SuSiE analysis (Table S10).

3. Novel Loci: The GWAS of KCPS2 identified a large number of novel loci (616). However, I wonder if this high number is due to the comparison being limited to loci reported by Open Targets. If significant loci from other East Asian biobanks were also considered as known, how many of these variants would still be classified as novel?

Thank you for this comment. Although the breadth of the GWAS literature and rapid pace of its expansion makes it challenging to be definitive, we were able to expand our list of known associations and refine our definition of novel loci. We incorporated additional filters from previous studies, including those from other East Asian biobanks such as KoGES, BBJ, and TWB. Specifically, we have made the following updates:

1. **Extended flanking region:** We now extend the loci by adding a ± 500 kb flanking region to ensure a more conservative identification of novel loci. Previously, we focused only on the loci themselves without including surrounding regions.
2. **Inclusion of recent GWAS studies:** We have added two recent thyroid-stimulating hormone GWAS studies (Sternborg et al., Nat Comm 2024, and William et al., Nat Comm 2023) that were not included in either the Open Targets database or the GWAS Catalog API used initially.
3. **Incorporation of East Asian biobanks:** We now include associations reported by recent East Asian biobank studies, such as Nam et al., Cell Genom 2024 for KoGES and Chen et al., Cell Genom 2024 for TWB. Associations reported by Sakaue et al., Nat Gen 2021 for BBJ were already included in the Open Targets database we used.

We have also added **Supplementary Figure 8**, which provides a flow chart illustrating the updated process for identifying novel loci. With these changes, we now report 301 novel loci, a 49% reduction from the 616 originally reported.

Supplementary Figure 8

Additionally, among these novel loci, how many can be replicated in other East Asian biobanks?

Thank you for this important question. To address it, we examined the KCPS2 novel lead SNPs in other East Asian biobanks (KoGES, BBJ, TWB) and counted the number of SNPs that were replicated. We defined replication as having the same direction of effect and a nominal significance (p -value < 0.05) in these other East Asian biobanks. Additionally, we counted the number of the novel loci reaching genome-wide significance in a meta-analysis of these biobanks.

Across 21 traits available in all five biobanks, 56%, 16% and 41% of all genome-wide significant loci in the KCPS2 replicated in KoGES, BBJ, and TWB, respectively. Of the novel genome-wide significant loci in KCPS2, 34%, 7%, and 22% of the novel loci were replicated in KoGES, BBJ, and TWB, respectively. Our analysis found that (with one exception) none of the novel loci reached genome-wide significance in any of the East Asian biobanks, reinforcing the novelty of our findings (column O-S in **Supplementary Table 12**). The one novel association that reached genome-wide significance in a second biobank was the association in the KCPS2 between a

SNP at 12:118399491 and fasting blood sugar (FBS) [EFO_0004465]. This SNP was found to be genome-wide significant in KoGES's glucose GWAS, which used a different EFO ID [EFO_0004468].

We also conducted a meta-analysis of the other three East Asian biobanks (KoGES, BBJ, and TWB) to further investigate replication. 48% of all loci genome-wide significant in the KCPS2 replicated in this meta-analysis of the other three biobank; 32% were genome-wide significant in the meta-analysis. 36% of the novel loci replicated; 4% were genome-wide significant (**Supplementary Table 12**).

We now refer to these replication results and cite Supplementary Table 12 on page 6.

minor typos

Ex. Line 520 : aAnthropometric
Corrected.

Reviewer #2 (Remarks to the Author):

This paper reported a quantitative trait GWAS study in the Korean Cancer Prevention Study-II (KCPS2) Biobank. This biobank contains 153,950 participants, and the study focused on 36 quantitative traits. They reported 616 novel genetic loci in KCPS2 and meta-analyzed them with Biobank Japan and Taiwan Biobank, etc. Overall, these results contribute to the study of the genetic architecture of complex traits and diseases within East Asia, which is currently scarce.

I have the following comments:

1. KCPS2 cohort characteristics need to be presented and commented. Ideally, it needs to be compared with other Korean and/or East Asian cohorts.

We have now added the characteristics of the KCPS2 cohort, along with comparisons to other biobanks, in **Supplementary Table 13**. Additionally, we have provided commentary in the main text.

To summarize, the **Supplementary Table 13** shown below highlights the key characteristics across five biobanks, including four East Asian biobanks (three population-based and one hospital-based) and one European biobank (UKB).

<**Supplementary Table 13**>

Biobank	Acronym	Recruitment country	Sample ascertainment	Sample size	Sources	GWAS method	Baseline mean age (yr)	Female (%)
Korean Cancer Prevention Study-II Biobank	KCPS2	Korea	Population-based	153,950	Individual-level data	Linear mixed model (LMM) in SAIGE	42	40%
Korean Genome and Epidemiology Study	KoGES	Korea	Population-based	72,000	Nam et al., Cell Genom 2022	LMM in SAIGE	54	64%
Biobank Japan	BBJ	Japan	Hospital-based	179,000	Sakaue et al., Nat Gen 2021	LMM in BOLT	63	46%
Taiwan Biobank	TWB	Taiwan	Population-based	102,900	Chen et al., Cell Genom 2023	LMM in REGENIE	50	64%
UK Biobank	UKB	UK	Population-based	420,000*	Karczewski et al., medRxiv 2024	LMM in SAIGE	57	53%

*the UKB analyses were restricted to the EUR genetic ancestry group

2. How does this biobank compare with other Korean biobanks mentioned in the paper, e.g., KoGES? Does the sample overlap with each other?

We do not have access to individual-level data for KoGES, so could not directly identify and count any overlapping participants. However, multiple lines of evidence suggest the overlap between KCPS2, KoGES, and the other East Asian biobanks is negligible. KoGES and KCPS2 cohorts were recruited at different times and regions. Previously published geographical information also gives reassurance that the overlap between BBJ, TWB, and UKB is expected to be small.

1) Contextual evidence:

KCPS2 participants were predominantly (90%) recruited from Seoul and Gyeonggi Province between 2004 and 2013, whereas KoGES participants were recruited from across South Korea between 2001 and 2013, as shown in the figure below. KCPS2 participants were recruited from patients attending 18 specific health promotion centers, while KoGES was an amalgamation of multiple studies using different ascertainment schemes (e.g. the Ansan and Ansong study is a community-based cohort randomly recruited from the Ansan [urban] and Ansong [rural] regions outside Seoul via mail, telephone and home visits; the Cardiovascular Disease Association Study was restricted to rural counties; and the KoGES Immigrant and Emigrant studies were restricted to Immigrants and Emigrants, respectively [the latter residing in Japan]). The limited

overlap in recruitment time, place and mechanism suggests the sample overlap between KCPS2 and KoGES is likely small.

KCPS2	KoGES
[REDACTED]	[REDACTED]
Jee et al., IJE 2017	Kim et al., IJE 2016

2) Empirical evidence using LDSC cross-biobank genetic correlation intercept:

We also compared the genetic correlation intercepts between biobanks using LDSC, which can capture potential sample overlap. The table below shows the mean intercept estimates across 19 traits for comparisons between KCPS2 and other biobanks (the estimates were extracted from **Supplementary Table 9**). The intercept for KCPS2-KoGES is slightly above 0 (0.0516), but not substantially different from other non-Korean biobanks. These results suggest that if there is any overlap between KCPS2 and KoGES, it would be very small and would not have a significant impact on our findings.

Mean gcov_intercept (19 traits)	Value
KCPS2-KoGES	0.0516
KCPS2-BBJ	0.046731579
KCPS2-TWB	0.035084211
KCPS2-UKB	0.043621053

3. I am also curious about the coverage of the overall Korean population. Are they mostly from big cities or rural areas, south or north? The South Korean population is 51 million, and KCPS2 covers only 0.2%. Are they enriched with people having cancer risk factors?

The KCPS2 cohort is not fully representative of the entire South Korean population, as the majority of participants were recruited from the Seoul and Gyeonggi regions. Approximately 90% of participants came from these regions, where about 40% of the South Korean population resides (around 19 million people). Therefore, the cohort may not provide generalizable estimates for the entire population, particularly for characteristics that vary geographically across the country.

KCPS2 is a population-based cohort study; participants were not enrolled on the basis of cancer risk factors (e.g. family history of cancer). Recruitment occurred during visits to participating health centers, including regular check-ups (33%), general health concerns (29%), and mandatory group examinations from work (16%). Long-term follow-up is conducted using unique linkages to routine nationwide medical examinations conducted at health promotion centers. These records include data on mortality and hospitalization. While cancer research is one focus of the study, KCPS2 was also established to investigate the determinants and long-term consequences of metabolic syndrome and other chronic diseases.

Despite the regional focus, the cohort's large sample size, comprehensive linkage to mortality and morbidity records, and availability of baseline blood samples make it well-suited to study a wide range of health behaviors, biological characteristics, and genetics. More detailed information on the cohort's coverage and baseline characteristics can be found in Jee et al., *IJE* 2017. We added a short description on the coverage of the overall Korean population to the methods section describing KCPS2.

4. How were the biomarkers QCed? Are they from the EHR? Were these measured once or multiple times?

Blood samples were collected from participants explicitly for KCPS2 at study baseline. Fasting blood samples were taken and serum and EDTA whole blood put into storage at -70C for future study. Serum was also used for immediate biomarker analysis. Glucose, total cholesterol, triglyceride, high-density lipoprotein cholesterol (HDL-c), low-density lipoprotein cholesterol (LDL-c) and other biomarkers were measured in the hospital laboratory by a COBAS INTEGRA 800 and a 7600 Analyzer (Hitachi, Tokyo, Japan). LDL-c was either directly measured or calculated based on total, HDL-c and triglyceride levels. Each laboratory had internal and external quality control procedures as required by the Korean Association of Laboratory Quality Control. Agreement for each biomedical marker across individual hospitals was high (correlation 0.96–0.99).

Participants may have attended for re-examinations, similar to the baseline assessment, every year. By the end of the baseline recruitment, 71,580 participants (46%) had had at least one such repeat examination (on average the first repeat visit was just over a year after their baseline assessment), 20,694 (13%) had had at least two repeat examinations, and 7,033 (5%)

had had at least three repeat examinations. At each examination, participants complete a questionnaire and provide a fasting blood sample (with the same procedures as at recruitment). In this study, however, we only used baseline measurements for all biomarkers.

More information on “Blood sampling and assays” and “Re-surveys” can be found in Jee et al., *IJE* 2017.

5. Heritability was estimated to be lower than usual. Is it due to the use of summary statistics from GWAS? What if individual data is used to estimate heritability?

The reviewer touches on an important point, namely that different methods for estimating heritability have different estimands and different analytic properties. The heritability estimated from twin studies e.g. need not be the same as the heritability estimated from GWAS data (a.k.a. “SNP heritability”)--and different methods for estimating SNP heritability (using individual level data or GWAS summary statistics) can lead to different estimates. Here we have focused on estimates of SNP heritability based on GWAS summary statistics (i.e. LDSC regression and SBayesS) in order to have an apples-to-apples comparison of genetic architectures across biobanks and traits. Our focus is not on the absolute value of SNP heritability, but the relative performance (are heritability estimates similar across biobanks? how do heritability estimates vary across traits?).

We agree that it is expected for SNP-heritability to be lower than twin heritability, as the latter captures more of the genetic variation, including non-additive genetic effects, while SNP-heritability often reflects only part of the total heritability, leading to the phenomenon of “missing heritability.” Furthermore, as shown in Figure 4, our SNP-heritability estimates are consistent with those reported in other East Asian biobanks, although they may vary due to differences in study design, sample size, and environmental factors.

6. The novel rate of thyroid-stimulating hormones is high. Any reason for this? Is there any gender difference for this, given thyroid dysfunction is more common in females?

We appreciate the reviewer’s thoughtful question. We believe the high novel rate of thyroid-stimulating hormone (TSH) variants in KCPS2 may be due to the fact that TSH is not typically measured at baseline in other biobanks, including KoGES, BBJ, TWB, and UKB. This lack of comparative data likely contributes to the identification of novel loci in our analysis.

Although the proportion of females in KCPS2 is higher than that of males (with females comprising 60% of the cohort), we adjusted for sex in our GWAS of TSH, as we did for all GWAS analyses. We observed significantly higher TSH levels in females compared to males (female: 2.1 vs. male: 1.8 mIU/L; p-value < 2.2e-16). This suggests that there could be sex-specific differences in how genetic variants influence TSH levels, though further exploration of this topic is beyond the scope of the current paper.

7. How to interpret $r_g > 0$, but r_p is zero or very close to zero?

This is a good question. We believe that many of these apparent discrepancies are the result of statistical or mathematical artifacts and idiosyncrasies. Many instances where genetic correlations (r_g) are positive but phenotypic correlations (r_p) (e.g., shown in below table) are close to zero are likely due to chance, where r_g may be overestimated and r_p underestimated, often referred to as the “winners’ curse,” rather than having a clear biological interpretation.

p1	p2	rg	rg_fdr	rp	rp_fdr	fdr_cat
ALB (albumin)	PLT (platelet)	0.1279	0.016448	-0.00907	0.262112	Only r_g significant
EOS (eosinophil)	ALCO_AMOUNT (alcohol intake)	0.2145	0.026871	-0.00103	0.899777	Only r_g significant

Given that phenotype data can be noisy, it can be challenging to discern the underlying cause of the discrepancy between r_g and r_p ($r_g * r_p < 0$). This issue may be particularly complex for trait pairs that are related by ratio, as these relationships can introduce additional complexities. Notably, many of these pairs involve hematological traits, such as i) mean corpuscular hemoglobin (MCH) and mean corpuscular hemoglobin concentration (MCHC), with $r_g = 0.60$ and $r_p = -0.23$ (both r_g and r_p significant) and ii) MCH and red blood cell (RBC), with $r_g = -0.55$ and $r_p = 0.009$ (only r_g significant). MCH and MCHC are calculated as $MCH = \text{hemoglobin} / \text{RBC}$ and $MCHC = \text{hemoglobin} / \text{hematocrit}$, respectively. The ratios are complex because the numerators and denominators may be influenced by different factors.

8. Why was the correlation with KCPS2 effect sizes greatest for the BBJ and not the other Korean cohort? It would be really helpful to compare the cohort characteristics...

We are happy to provide additional context in response to this question. First, as noted in our response to Reviewer 1, the correlations in effect sizes between KCPS2 and KoGES, TWB, and BBJ are actually quite similar (all 0.89), and we have updated the manuscript accordingly. We originally only presented the slopes from regressions of effect-size estimates from KCPS2 on effect-size estimates from the other biobanks; as noted in our response to Reviewer 1, differences in slopes are confounded by differences in estimate variances across studies.

Second, we have provided additional information describing study characteristics. KCPS2 and KoGES are population-based cohorts, while BBJ is a hospital-based cohort, for example. In response to the reviewer’s comment 1 as well, we have now included a detailed comparison of the cohort characteristics in Supplementary Table 13 and provided commentary in the main text.

To summarize, the **Supplementary Table 13** shown below highlights the key characteristics across five biobanks, including four East Asian biobanks (three population-based and one hospital-based) and one European biobank (UKB).

<Supplementary Table 13>

Biobank	Acronym	Recruitment country	Sample ascertainment	Sample size	Sources	GWAS method	Baseline mean age (yr)	Female (%)
Korean Cancer Prevention Study-II Biobank	KCPS2	Korea	Population-based	153,950	Individual-level data	Linear mixed model (LMM) in SAIGE	42	40%
Korean Genome and Epidemiology Study Biobank	KoGES	Korea	Population-based	72,000	Nam et al., Cell Genom 2022	LMM in SAIGE	54	64%
Biobank Japan	BBJ	Japan	Hospital-based	179,000	Sakaue et al., Nat Gen 2021	LMM in BOLT	63	46%
Taiwan Biobank	TWB	Taiwan	Population-based	102,900	Chen et al., Cell Genom 2023	LMM in REGENIE	50	64%
UK Biobank	UKB	UK	Population-based	420,000*	Karczewski et al., medRxiv 2024	LMM in SAIGE	57	53%

*the UKB analyses were restricted to the EUR genetic ancestry group

9. Why meta-analyze with UKB? Together with all the East Asian cohorts?

We greatly appreciate the reviewer's thoughtful question. We chose to meta-analyze the East Asian biobanks with the European population from UK Biobank (UKB) for two key reasons. First, there is substantial evidence that genetic loci discovered in one population, such as Europeans, often replicate in other populations, such as African or East Asian populations, as seen in previous studies. By leveraging the shared biology across populations, meta-analyzing with UKB helps increase statistical power, enabling us to better resolve which variants are associated with the traits of interest and are more likely to be causal.

Additionally, UKB offers a broader range of phenotypes with greater coverage compared to some East Asian biobanks, which further strengthens the robustness of the meta-analysis.

Reviewer #3 (Remarks to the Author):

In this manuscript, the authors performed GWAS on 36 quantitative traits in KCPS2, a large Korean Biobank with 153,950 individuals. They identified 616 novel genetic loci in KCPS2 and 3,524 loci in a meta-analysis with the Korean Genome and Epidemiology Study (KoGES), Biobank Japan (BBJ), Taiwan Biobank (TWB), and the UK Biobank (UKB). They described differences in the genetic architecture between East Asian and European populations, and highlighted East Asian specific associations. The manuscript is well structured and clearly written, and I greatly appreciate the effort in improving the population diversity of GWAS. I have a few questions and suggestions for the authors, which I listed below.

1. I think a major limitation is that the genetic architecture analysis was not performed in KoGES, given that it was the only other large Korean study included in this manuscript. The authors suggested that summary statistics from linear regression models were not available, but it was unclear how sensitive their analyses would be using summary statistics from linear mixed models instead (i.e., how did these summary statistics compare with each other). For example, they estimated cross-trait genetic correlations in KCPS2 using LDSC and summary statistics from linear regression models, but their cross-biobank genetic correlations between KCPS2 and KoGES were likely estimated using LDSC and summary statistics from linear mixed models (at least in KoGES). Many GWAS summary statistics-based methods were originally derived in the linear regression framework, but they have been applied to GWAS summary statistics from more complex models such as linear mixed models. On the other hand, the sample size in KoGES (72K) was not dramatically smaller than TWB (102K). Some further justifications for leaving out KoGES in this analysis would be helpful.

After a careful re-evaluation of our analysis, we identified a few issues that may have led to unstable results in the initial SBayesS estimates for KoGES. We have since revised our analysis pipeline to address these concerns.

For our updated analysis, we made the following corrections:

1. We removed 7,998 related samples from the LD matrix construction in KCPS2, which is used for SBayesS analysis across KCPS2, TWB, and KoGES.
2. We removed related samples from the GWAS summary statistics in KCPS2, which is related to Reviewer 3's comment 2.
3. We included the `--genmap-n 504` option to correctly specify the reference panel sample size of the five East Asian populations from the 1000 Genomes Project. Previously, the default value of 183 (referring to the CEU cohort in the 1000 Genomes Project) was used, which was incorrect.
4. We ensured the correct specification of the sample size column (N) in KoGES, which had been misrepresented in our earlier analysis.

After making these corrections, we found no significant or unreliable differences in the genetic architecture parameters between KoGES and the other biobanks. Consequently, we have now included the KoGES results in the main figures and updated the text accordingly.

We appreciate the reviewer’s suggestion, as it has helped us improve the analysis and provide a more comprehensive comparison across all biobanks.

2. The validity of simple linear regression needs to be justified, as it could lead to spurious association findings in the presence of relatedness. For UKB, GWAS summary statistics from linear regressions (lines 592-594 on page 29) are likely invalid, given that the sample size (420K) suggested that related individuals had been included.

Thanks for raising this point; we are happy to provide clarifications. For the genetic architecture analyses using SBayesS, the UKB and BBJ estimates were extracted from Wang et al., 2022, where related individuals had been removed. As a result, the sample size used for the simple linear regression model in UKB (N = 361,144) is smaller than the full sample size (N = 420K) used for the linear mixed model. We have clarified in the Methods section that the SBayesS estimates were derived from unrelated individuals in both UKB and BBJ, as follows:

“For BBJ and UKB, we used the previously reported genetic architecture parameter estimates,65 which were constructed using GWAS summary statistics generated from linear regression models and in-sample LD for the corresponding unrelated population.”

Regarding KCPS2, we acknowledge the reviewer's concern about the potential for spurious associations due to relatedness. In response, we have rerun the SBayesS analysis, restricting the estimation of genetic architecture parameters to unrelated participants in KCPS2. The results have been updated in Figure 4, and the Methods section has been revised to reflect this change, as follows:

“For a fair comparison of these parameters between KCPS2, BBJ, TWB, and UKB, we applied SbayesS to GWAS summary statistics generated from linear regression models in unrelated KCPS2 and TWB, instead of linear mixed models.”

3. The novel loci in KCPS2 tend to be more common than in 1000G EUR samples (lines 90-92 on page 5). What about the MAF of these novel loci in other East Asian samples (e.g., 1000G EAS, or BBJ, KoGES, TWB)?

Thank you for suggesting we provide this additional context. The median MAF of the novel loci in KCPS2 (0.207) is more similar to that in the 1000 Genomes East Asian (1KG EAS) population (median EAS MAF: 0.202, paired t-test $P = 3.8e-4$) than that in the 1KG EUR population (median EUR MAF: 0.118; paired t test $P = 2.2e-16$).

We now include the comparison of MAF between novel loci in KCPS2 and 1KG EAS in the main text and Supplementary Figure 1 as follows:

The novel loci tend to be more common in KCPS2 than in 1000 Genome phase 3 EUR samples (median KCPS2 minor allele frequencies [MAF]: 0.207 vs. median EUR MAF: 0.118; paired t test $P = 2.2e-16$ vs. median EAS MAF: 0.202; paired t-test $P = 0.3.8e-4$). (Figure S1).

Additionally, we have examined the median MAF of these novel loci for the 21 traits available across all biobanks in other East Asian biobanks, as summarized below. Please note the median MAF of KCPS2 was 0.247 across these 21 traits, which is different from the median MAF of KCPS2 across 36 traits in KCPS2 (0.207).

MAF.KCPS2	MAF.KoGES	MAF.BBJ	MAF.TWB	MAF.UKB
0.246763	0.2400122	0.235364	0.2692	0.20385

These results suggest that the novel loci are similarly common across various East Asian cohorts, further supporting the robustness of our findings. We have added these results to the manuscript for clarity.

4. How were the lead variants defined in Figure 3? For one locus, it is possible that the lead variant in KCPS2 might be different from the lead variant other biobanks (or the lead variant from meta-analysis). If the lead variants were different across studies, were they in LD with each other? How about their MAF's?

The lead variants in Figure 3 were defined based on the genome-wide significant loci identified from the meta-analysis. At each locus (defined as the 1 Mb region 500kb upstream and downstream of the lead variant), we filtered genome-wide significant variants by removing those with LD $r^2 > 0.01$ in the 1000 Genomes ALL reference panel. We recognize that this is a first-pass, approximate attempt to identify statistically independent associations at each locus. Additional fine mapping analyses accounting for different in-sample linkage disequilibrium patterns across the contributing biobanks and possible differences in effect sizes are beyond the scope of this paper.

We examined the MAF of these variants across studies, which can be found in text and Supplementary Figure 4.

The lead variants from genome-wide significant loci identified in the meta-analysis had in general similar study-specific MAF in East Asian populations (KCPS2 [median MAF=0.276], KoGES [median MAF=0.276], and BBJ [median MAF=0.278], and TWB [median MAF=0.28]) compared to European ancestry populations in UKB (median MAF=0.26).

Also I find it a little difficult to interpret the results that the correlation between KCPS2 and BBJ effect size estimates was greater than the correlation between KCPS2 and KoGES effect size estimates (lines 144-145 on page 8), given that both KCPS2 and KoGES samples were Koreans. Was that caused by the different traits analyzed across these studies, and/or different lead variants across studies?

We are happy to provide additional context in response to this question. First, as noted in our response to Reviewer 1 and 2, the correlations in effect sizes between KCPS2 and KoGES, TWB, and BBJ are actually quite similar (all 0.89), and we have updated the manuscript accordingly. We originally only presented the slopes from regressions of effect-size estimates from KCPS2 on effect-size estimates from the other biobanks; as noted in our response to Reviewer 1, differences in slopes are confounded by differences in estimate variances across studies.

Second, we have provided additional information describing study characteristics. KCPS2 and KoGES are population-based cohorts, while BBJ is a hospital-based cohort, for example. In response to the reviewer's comment 1 as well, we have now included a detailed comparison of the cohort characteristics in Supplementary Table 13 and provided commentary in the main text.

To summarize, the **Supplementary Table 13** shown below highlights the key characteristics across five biobanks, including four East Asian biobanks (three population-based and one hospital-based) and one European biobank (UKB).

<Supplementary Table 13>

Biobank	Acronym	Recruitment country	Sample ascertainment	Sample size	Sources	GWAS method	Baseline mean age (yr)	Female (%)
Korean Cancer Prevention Study-II Biobank	KCPS2	Korea	Population-based	153,950	Individual-level data	Linear mixed model (LMM) in SAIGE	42	40%
Korean Genome and Epidemiology Study	KoGES	Korea	Population-based	72,000	Nam et al., Cell Genom 2022	LMM in SAIGE	54	64%
Biobank Japan	BBJ	Japan	Hospital-based	179,000	Sakaue et al., Nat Gen 2021	LMM in BOLT	63	46%
Taiwan Biobank	TWB	Taiwan	Population-based	102,900	Chen et al., Cell Genom 2023	LMM in REGENIE	50	64%
UK Biobank	UKB	UK	Population-based	420,000*	Karczewski et al., medRxiv 2024	LMM in SAIGE	57	53%

*the UKB analyses were restricted to the EUR genetic ancestry group

5. I am not sure if median SNP-heritability would be a good comparison between KCPS2 and BBJ since the traits analyzed were different across studies (line 171 on page 10). Some traits might have a higher/lower heritability but were unavailable in some studies. Was the difference between 0.26 and 0.25 statistically meaningful? The benchmark could be more rigorous if performed on the same traits across all studies.

We sincerely thank the reviewer for this valuable suggestion. In response, we have updated our analysis to ensure that, when comparing the median SNP-heritability of trait categories between

two studies, we restricted the comparison to the same list of traits within each category for that specific biobank pair. To assess statistical significance, we performed a Wilcoxon signed-rank test, but no significant differences were observed ($p > 0.05$), likely due to the limited number of traits being compared (number of paired comparisons range from 1 to 8). We have now provided the Wilcoxon signed-rank test results in Supplementary Table 14 and updated the Results and Method section as follows:

Results:

However, these differences [for a given trait category] were not statistically significant (Wilcoxon signed-rank test $p > 0.05$), likely due to the limited number of traits being compared (number of paired comparisons range from 1 to 8) (Table S15).

Methods:

When comparing the median SNP-heritability of trait categories between two studies, we restricted the comparison to the same list of traits within each category for that specific biobank pair and performed a Wilcoxon signed-rank test to assess statistical significance.

Also I am not sure if correlation of heritability estimates (lines 175-178 on page 10) is the best measure, given limited numbers of traits (and different traits analyzed across studies). Heritability estimates could be highly correlated, but still quite different between two studies (e.g., 0.6, 0.4, 0.2 for three traits in one study, and 0.3, 0.2, 0.1 in another study).

We fully agree with the reviewer that correlations may vary depending on the traits available in each study, which could lead to inconsistent comparisons. To address this, we have added correlations based on the 8 traits that are available across all five studies, and we have included this information in the footnote of Figure 4.

For the 8 traits available in all five studies, we observed high correlations of heritability between KCPS2 and the other biobanks: KoGES (Pearson correlation $r=0.99$, 95% confidence interval [CI]: 0.97-1.00), BBJ ($r=0.93$, 95% CI: 0.64-0.99), TWB ($r=0.93$, 95% CI: 0.65-0.99), and UKB ($r=0.97$, 95% CI: 0.82-0.99).

Was the reported lower correlation with TWB (line 176 on page 10) statistically significant? The confidence intervals were very wide (possibly due to the limited numbers of traits) and they appeared to overlap with each other.

The correlation between KCPS2 and the TWB heritabilities for the 8 overlapping traits across all studies was statistically significantly different from 0 ($r=0.64$, 95% CI: 0.31–0.83, $p=9.62E-04$), but we could not say with confidence that the KCPS2-TWB correlation was statistically significantly lower than the correlation with other biobanks, due to the large confidence intervals. We have now added the statistical significance and confidence intervals to the text.

..which led to a low correlation between the heritability estimates of KCPS2 and TWB ($r=0.64$, 95% CI: 0.31-0.83, $p=9.62E-04$).

6. I agree with the authors that using in-sample LD is critical, since LD scores could be different in KCSP2 and TWB. It might be better to move TWB in-sample LD results to Figure 4, and TWB results using LD scores based on 50K KCPS2 samples to Supplementary Figures.

We thank the reviewer for this valuable suggestion. In response, we have now moved the TWB in-sample LD results to Figure 4, and the TWB results using LD scores based on the 50K KCPS2 samples to Supplementary Figure 6. This adjustment is also clarified in the footnote of Figure 4.

For heritability in TWB shown in (d), we used the heritability estimates reported by Chen and colleagues.

7. The authors mentioned that phenotypes were rank normal transformed (line 542 on page 27). Does that mean all the reported beta's were effect size estimates on the transformed scale (i.e., not on the raw phenotypes)? Could the authors confirm that the same rank normal transformation was performed for all these phenotypes in the KoGES, BBJ, TWB and UKB GWAS summary statistics? Otherwise the inverse-variance-weighted fixed-effect meta-analysis should not be performed if the effect size estimates were on different scales.

We confirm that the same rank normal transformation was performed for all these phenotypes in the KoGES, BBJ, TWB and UKB GWAS summary statistics.

Minor:

1. I am not sure if the direction from "meta-analysis" to GWAS is accurate in Figure 1. I thought GWAS (in KCPS2) should precede meta-analysis?

Corrected.

2. It would be great to improve the alignment in Figure 2a. Currently there is a slight shift due to the extra space between trait groups on the left panel (but not on the right panel), so they do not align well horizontally. The font size was too small for "Thyroid hormone" and "Tumor marker" on the left panel.

Corrected.

3. After revising Figure 2a, it would be great to also format Figure S3 in the same style.

Corrected.

4. Gene names in Figure 2b could be better presented if they are annotated horizontally near the dots. Vertical lines look a little confusing when they stack together.

Corrected.

5. In Figure 2c, the INSULIN-CEA pair was shown in blue (only r_p significant) but the point estimate was very close to 0 for r_p and largely negative for r_g . It would be nice to double check. In addition, if the slope between pairwise r_g and r_p was estimated from a linear regression model, I think the standard error estimate (line 116 on page 7) might not be very

informative, since the independence, normality and homoscedasticity assumptions were likely violated for r_g . Also, which FDR threshold was used?

Thank you for your detailed review. We have double checked the INSULIN-CEA pair, and the Figure 2c correctly captures the correlation estimates, as shown in below table. We believe that instances where genetic correlations (r_g) are largely negative while phenotypic correlations (r_p) are close to zero are likely due to chance, where r_g may be overestimated and r_p underestimated, often referred to as the “winners’ curse,” rather than reflecting a clear biological interpretation.

p1	p2	rg	rg_fdr	rp	rp_fdr	fdr_cat
INSULIN	CEA	-0.3664	0.1091444516	-0.01783363119	0.026136	Only r_p significant

We agree with the reviewer that the standard error estimate for the slope between pairwise r_g and r_p estimated from a linear regression model might not be very informative. Consequently, we have removed this estimate from the text.

Regarding the FDR threshold, we used an $FDR < 0.05$ for significance. We have clarified this in Figure 2 caption as follows:

Significant rg and rp after false discovery rate ($FDR < 0.05$) correction is indicated by purple if both rg and rp were significant, red if only rg was significant, blue if only rp was significant, and gray if neither was significant.

6. Since the coding allele in GWAS might be arbitrary, it would be better to present Figure 3b and Figure S4 as plots versus the MAF (instead of AF) on the x axis.

Corrected

7. Visualization could be improved in Figure 4a. The current overlay and alignment make it difficult to tell which dots came from which traits.

Corrected.

8. It is unclear whether $\text{neglog}P$ in Figure 5 refers to natural log or log base 10. For p-values, I think log base 10 would be more informative. In Figure 5b, trait labels were truncated at the top. If coloc.pp4 was all equal to 1 for these 7 traits, it could be removed from this figure and described only in the text (e.g., “posterior probability of colocalization at the specified region”, instead of “PP4”).

Log base 10 was used. Corrected.

9. In the Discussion, when presenting beta or odds ratio for a variant, it would be great to specify which allele the authors were referring to. Flipping the alleles could change the direction of effect.

Corrected.

10. Typos: “wWe” in line 556 on page 27; “Zendo” in line 631 on page 31.
Corrected.

Reviewer #4 (Remarks to the Author):

REVIEWERS' COMMENTS

We thank the reviewers for their careful and thoughtful reviews. In response, we have revised Figure 5b. The reviewers' comments have helped us improve the manuscript.

Reviewer #1 (Remarks to the Author):

The authors addressed all of my comments. I don't have any additional ones.

Reviewer #1 (Remarks on code availability):

I checked the GitHub page, and it appears that all the relevant codes are available.

Reviewer #3 (Remarks to the Author):

I appreciate the authors' efforts in addressing my previous comments. The revised manuscript has been improved in its clarity. I only have two minor suggestions for them to consider in Figure 5b:

1. The authors have removed coloc.PP4 in the figure since it was equal to 1 for all seven traits. I would suggest replacing "Coloc.PP4 represents the posterior probability of colocalization at the specified region. All seven traits shown here had coloc.PP4=1 with alcohol intake" by something like "All seven traits shown here had a posterior probability of colocalization at the specified region of 1 with alcohol intake."

Thank you for this observation. We have now replaced the sentence by "All seven traits shown here had a posterior probability of colocalization at the specified region of 1 with alcohol intake." in Figure 5b.

2. I would suggest increasing the width of the 95% CS legend boxes so that they don't have a second row to overlay with the lines of dots near PIP = 0.0 for GGT and DBP.

Thank you for this comment. We now increased the width of the 95% CS legend boxes in Figure 5b.

Reviewer #4 (Remarks to the Author):
